# Survey of Deep Learning Accelerators for Edge and Emerging Computing

Shahanur Alam [1,*], Chris Yakopcic [1], Qing Wu [2], Mark Barnell [2], Simon Khan [2] and Tarek M. Taha [1,*]

[1] Department of Electrical and Computer Engineering, University of Dayton, Dayton, OH 45469, USA; cyakopcic1@udayton.edu

[2] Information Directorate, Air Force Research Laboratory, Rome, NY 13411, USA; qing.wu.2@us.af.mil (Q.W.); mark.barnell.1@us.af.mil (M.B.); simon.khan@us.af.mil (S.K.)

\* Correspondence: alamm8@udayton.edu (S.A.); tarek.taha@udayton.edu (T.M.T.)

**Abstract:** The unprecedented progress in artificial intelligence (AI), particularly in deep learning algorithms with ubiquitous internet connected smart devices, has created a high demand for AI computing on the edge devices. This review studied commercially available edge processors, and the processors that are still in industrial research stages. We categorized state-of-the-art edge processors based on the underlying architecture, such as dataflow, neuromorphic, and processing in-memory (PIM) architecture. The processors are analyzed based on their performance, chip area, energy efficiency, and application domains. The supported programming frameworks, model compression, data precision, and the CMOS fabrication process technology are discussed. Currently, most commercial edge processors utilize dataflow architectures. However, emerging non-von Neumann computing architectures have attracted the attention of the industry in recent years. Neuromorphic processors are highly efficient for performing computation with fewer synaptic operations, and several neuromorphic processors offer online training for secured and personalized AI applications. This review found that the PIM processors show significant energy efficiency and consume less power compared to dataflow and neuromorphic processors. A future direction of the industry could be to implement state-of-the-art deep learning algorithms in emerging non-von Neumann computing paradigms for low-power computing on edge devices.

**Keywords:** AI accelerator; AI frameworks; deep learning; edge computing; low-power applications; quantization; PIM or CIM computing; neuromorphic computing

## 1. Introduction

Artificial intelligence, and in particular deep learning, is becoming increasingly popular in edge devices and systems. Deep learning algorithms require significant numbers of computations ranging from a few million to billions of operations based on the depth of the deep neural network (DNN) models; thus, there is an urgent need to process them efficiently. As shown in Figure 1, two possible approaches for processing deep learning inference on edge devices are carried out directly on the device using highly efficient processors, fog, or cloud computing. A key benefit of fog/cloud-based processing is that large, complex models can be run without overburdening the edge device. The drawbacks of this approach are the need for a reliable communications channel, communications cost, communications delay, and potential loss of privacy.

In situations where a rapid response is needed, privacy is paramount; a reliable communications channel may not always be available, and processing of the deep learning network on the edge device or system may be the only option [1–3]. As a result, a large amount of academic and industrial research is being carried out to develop efficient deep learning edge processors [3]. Several companies have already announced or have started selling such processors. This paper provides details on these commercial deep learning edge

processors and compares their performance based on manufacturer-provided information. Additionally, the paper delves into the frameworks and applications related to these processors. The scope of edge computing includes end devices and edge nodes [4]. End devices include smartphones, wearables, autonomous cars, gadgets, and many more. Edge nodes are switches, routers, micro data centers, and servers deployed at the edge [5,6]. Table 1 lists some of the key characteristics of edge deep learning processors that are considered in this paper.

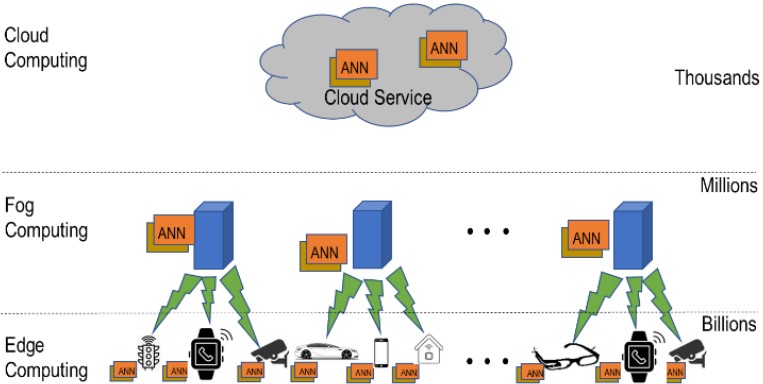

**Figure 1.** Illustration of edge computing with cloud interconnection.

**Table 1.** Brief scope of this paper.

| Architecture | Precision | Process (nm) | Metrics | Frameworks | Algorithm/Models | Applications |
|---|---|---|---|---|---|---|
| GPU<br>TPU<br>Neuromorphic<br>PIM<br>SoC<br>ASIC | FP-8,16,32<br>BF-16<br>INT-1,2,4,8,16 | 4<br>5<br>7<br>10<br>14<br>16<br>20<br>22<br>28<br>40 | Area<br>Power<br>Throughput<br>Energy<br>Efficiency | Tensorflow (TF)<br>TF Lite<br>Caffe2<br>Pytorch<br>MXNet<br>ONNX<br>MetaTF<br>Lava<br>Nengo<br>OpenCV<br>DarkNet | SNN<br>MLP<br>CNN<br>VGG<br>ResNet<br>YOLO<br>Inception<br>MobileNet<br>RNN<br>GRU<br>BERT<br>LSTM | Defense<br>Healthcare<br>Cyber Security<br>Vehicle<br>Smartphone<br>Transportation<br>Robotics<br>Education<br>UAV Drones<br>Communication<br>Industry<br>Traffic Control |

There are multiple types of AI accelerators enabling DNN computing: central processing units (CPUs), graphics processing units (GPUs), tensor processing units (TPUs), application-specific integrated circuits (ASICs), system on-chip (SoC), processing in-memory (PIM), and neuromorphic processors. ASICs, SoC, TPUs, PIM, and neuromorphic systems are mainly targeted at low-power AI applications in edge and IoT devices. Google introduced different versions of the TPU that are used in the Google cloud and the edge for training and inference [7]. Neuromorphic processors are non-von Neumann computing systems that mimic human cognitive information-processing systems. They generally utilize spiking neural networks (SNNs) for processing [8–15]. Several tech companies, including Intel and IBM [8–10], have developed brain-inspired neuromorphic processors for edge applications. PIM is another non-von Neumann computing paradigm that eliminates the data transfer bottleneck by having the computation take place inside a memory array in a highly parallel fashion [16–20].

PIM technology reduces data movement and latency compared to traditional architectures and makes computations significantly more efficient. Edge processors usually perform inference with highly optimized DNN models. The models are often compressed to reduce the number of computations, and the weight precision is usually quantized from the floating point (FP) format normally used in training. The quantized integer (INT) and

brain-float (BF) are used in inference processors. Typically, INT4, INT8, INT16, FP16, or BF16 numerical precision is used in the inference processor. However, recently released processors from multiple startups can compute with very low precision while trading off accuracy to some extent [21].

The current trend in computing technology is to enable data movement faster for higher speed and more efficient computing. To achieve this, AI edge processors need some essential prerequisites: lower energy consumption, smaller area, and higher performance. Neuromorphic and PIM processors are becoming more popular for their higher energy efficiency and lower latency [9,10,19,20]. However, a single edge processor usually does not support all types of DNN networks and frameworks. There are multiple types of DNN models, and each usually excels at particular application domains. For example, recurrent neural networks (RNNs), long short-term-memory (LSTM), and gated recurrent units (GRUs) are suitable for natural language processing [22–28], but convolutional neural networks (CNNs), residual neural network (ResNet), and visual geometry group (VGG) networks are better for detection and classification [29–31].

The CMOS technology node used for fabricating each device has a significant impact on its area, energy consumption, and speed. TSMC currently uses 3 nm extreme ultraviolet (UV) technology for the Apple A17 processor [32]. TSMC is currently aspiring to develop 2 nm technology by 2025 for higher performance and highly energy-efficient AI computing processors [33]. Samsung's smartphone processor Exynos 2200, developed with 4 nm technology, is on the market [34]. Intel utilized its Intel-4/7 nm technology for its Loihi 2 neuromorphic processor [9].

This article provides a comprehensive review of commercial deep learning edge processors. Over 100 edge processors are listed along with their key specifications. We believe this is the most comprehensive technical analysis at present. The main contributions of this review are as follows:

1.  It provides a comprehensive and easy-to-follow description of state-of-the-art edge devices and their underlying architecture.
2.  It reviews the supported programming frameworks of the processors and general model compression techniques to enable edge computing.
3.  It analyzes the technical details of the processors for edge computing and provides charts on hardware parameters.

This paper is arranged as follows: Section 2 describes key deep learning algorithms very briefly. Section 3 describes model compression techniques commonly used to optimize deep learning networks for edge applications. Section 4 discusses the frameworks available for deep learning AI applications. Section 5 describes the frameworks for developing AI applications on SNN processors. The processors are reviewed briefly in Section 6. Section 7 discusses the data on the processors and performs a comparative analysis. A brief summary of this review study is presented in Section 8.

## 2. Deep Learning Algorithms in Edge Application

Deep learning (DL) is a subset of AI and machine learning. It consists of multilayered artificial neural network architectures that optimize the network learning parameters to recognize patterns and sequences for numerous applications. The networks can be trained for specific tasks, such as speech recognition [35], image recognition [36,37], security [38], anomaly detection [39], and fault detection [40]. Deep learning algorithms can be classified into the following categories: supervised, semi-supervised, unsupervised, and deep reinforcement learning [41,42].

This study is focused on AI accelerators for edge/IoT applications. Supervised and semi-supervised DL categories are usually trained on high-performance computing systems and then deployed to edge devices. Supervised learning models utilize labeled data samples. These models usually extract key features from incoming data samples and use the features to classify the sample. One of the most popular categories of supervised DL networks is CNNs [42]. Some common CNN architectures include VGG [43], ResNet [44],

and GoogleNet [45]. Semi-supervised neural networks use a few labels to learn categories and could be generative models or time-based sequence learning models. The semi-supervised topologies include GAN, GRU, RNN, and LSTM. The internal layers of these NN models are composed of CNNs and fully connected network topologies. A number of edge processors support semi-supervised network models for automation applications. For example, DeepVision (now Kinara) introduced ARA-1 (2020) and ARA-2 (2022) [46], which target autonomous applications such as robotics, autonomous vehicles, smart tracking, and autonomous security systems. Kneron introduced KL720 in 2021, which supports semi-supervised network topologies for a wide range of applications [47]. In 2021, Syntiant released a new PIM AI processor for extreme edge applications, accommodating supervised and semi-supervised network topologies and supporting CNN, GRU, RNN, and LSTM topologies [20].

The computational complexity of DL models is a barrier to implementing these models for resource-constrained edge or IoT devices. For edge applications, the deep neural network should be designed in an optimized way that is equally efficient without losing a significant amount of accuracy. Common deep learning application areas for the edge include [48–55] image classification, object detection, object tracking, speech recognition, health care, and natural language processing (NLP). This section will discuss some lightweight DL models for edge applications that perform classification and object detection.

## 2.1. Classification

Classification is probably the most popular use of CNNs and is one of the key applications in the computer vision field [56–58]. While larger networks with higher accuracies are utilized in desktop and server systems, smaller and more highly efficient networks are typically used for edge applications.

SqueezeNet [59,60] utilizes a modified convolutional model that is split into squeeze and expand layers. Instead of $3 \times 3$ convolution operations seen in typical CNNs, a much simpler $1 \times 1$ convolution operation is used. SqueezeNet achieves AlexNet levels of accuracy with $50\times$ fewer network parameters [60]. Using model compression techniques, SqueezeNet can be compressed to 0.5 MB, which is about $510\times$ smaller than AlexNet.

MobileNet [61] was created by Google and is one of the most popular DL models for edge applications. MobileNet substitutes the traditional convolution operation with a more flexible and efficient depthwise separable operation, significantly reducing computational costs. The depthwise separable technique performs two operations: depthwise convolution and pointwise convolution. There are three available versions of MobileNet networks: MobileNet v1 [61], MobileNet v2 [62], and MobileNet v3 [63]. MobileNet v2 builds on MobileNet v1 by adding a linear bottleneck and an inverted residual block at the end. The latest MobileNet v3 utilizes NAS (neural architecture search) and NetAdapt to design a more accurate and efficient network architecture for inference applications [63].

ShuffleNet [64] utilizes group convolution and channel shuffle to reduce computation complexity. It increases accuracy by retraining with minimal computational power. There are two versions of ShuffleNet: ShuffleNet v1 and ShuffleNet v2 [64,65].

EfficientNet is a type of CNN with a corresponding scaling method that is able to find a balance between computational efficiency and performance. It can uniformly scale all the network dimensions, such as width, depth, and resolution, by using a compound coefficient [66]. The scaling method facilitates the development of a family of networks. Unlike other DL models, the EfficientNet model focuses not only on accuracy but also on the efficiency of the model.

## 2.2. Detection

Object detection is an important task in computer vision that identifies and localizes all the objects in an image. This application has a wide range of applications, including autonomous vehicles, smart cities, target tracking, and security systems [67]. The broad

range of object detection and DL network applications are discussed in [56,68]. DL networks for object detection can be categorized into two types: (i) single-stage (such as SSD, YOLO, and CenterNet) and (ii) two-stage (such as Fast/Faster RCNN). There are multiple criteria for choosing the right architecture for the edge application. Single-stage detectors are computationally more efficient than two-stage architectures, making them a better choice for edge applications. For example, YOLO v5 demonstrates better performance compared to Faster-RCNN-ResNet-50 [67].

### 2.3. Speech Recognition and Natural Language Processing

Speech recognition and natural language processing are becoming increasingly important applications of deep learning. Speech emotion and speech keyword recognition are the objectives of speech recognition. The process includes multiple state-of-the-art research fields, such as AI, pattern recognition, signal processing, and information theory. Apple's Siri and Google's Alexa illustrate the potential applications of speech recognition and manifest better computer–human interfacing. RNN-based neural networks and time delay DNN (TDNN) are popular choices for speech recognition [69]. Combined networks, such as TDNN-LSTM [70] or RNN-LSTM, are also popular choices for speech recognition [71].

Detailed analysis of deep neural networks for NLP can be found in [70,72]. Important applications of NLP are machine translation, named entity recognition, question-answering systems, sentiment analysis, spam detection, and image captioning. An early NLP model was sequence-to-sequence learning, based on RNNs. More recently, NLP was boosted by the advent of the transformer model, BERT [73]. BERT utilizes an attention mechanism that learns contextual relations between words [73]. Other state-of-the-art NLP models are GPT-2 [74], GPT-3 [75], GPT-4 [76], and the switch transformer [77]. However, these models run on HPC systems and are thus not compatible with edge devices. DeFormer [78], MobileBERT [79], and EdgeBERT [80] are some of the examples of NLP models targeted at edge devices. A more detailed discussion of NLP models for edge devices can be found in [81].

Syntiant [20] has recently been building tiny AI chips for voice and speech recognition and has attracted attention in the tech industry. Syntiant's Neural Decision Processors (NDPs) are certified by Amazon for use in Alexa-based devices [82]. Other voice recognition AI chips include NXP's i.MX8, i.MX9x [83–85] and M1076 from Mythic [86–88]. LightSpear 2803S from Gyrfalcon can be utilized for NLP [19,89]. IBM unveiled its NorthPole edge processor for NLP applications at the HotChips 2023 conference [90].

## 3. Model Compression

Unoptimized DL models contain considerable redundancy in parameters and are generally designed without consideration of power or latency. Lightweight and optimized DL models enable AI application on edge devices. Designing effective models for running on resource-constrained systems is challenging. DNN model compression techniques are utilized to convert unoptimized models to forms that are suitable for edge devices. Model compression techniques are studied extensively and discussed in [91–96]. The techniques include parameter pruning, quantization, low-rank factorization, compact filtering, and knowledge distillation. In this section, we will discuss some of the key model compression techniques.

### 3.1. Quantization

Quantization is a promising approach to optimizing DNN models for edge devices. Data quantization for edge AI has been studied extensively in [92–99]. Parameter quantization takes a DL model and compresses its parameters by changing the floating point weights to a lower precision to avoid costly floating point computations. As shown in Table 2, most edge inference engines support INT4, 8, or 16 precisions. Quantization techniques can be taken to the limit by developing binary neural networks (BNNs) [99]. A BNN uses a single bit to represent activations and reduces memory requirements. Leapmind

is a pioneer of low-precision computations in their edge processor, Efficiera [21]. It is an ultra-low-power edge processor and can perform AI computations with 1-bit weights and 2-bit activations.

**Table 2.** Commercial edge processors with operation technology, process technology, and numerical precision.

| Company | Latest Chip | Max Power (W) | Process (nm) | Area (mm²) | Precision INT/FP | Max Performance (TOPS) | Energy Efficiency (TOPS/W) | Architecture | Reference |
|---|---|---|---|---|---|---|---|---|---|
| Analog Devices | MAX78000 | 1 pJ/MAC | -- | 64 | 1, 2, 4, 8 | -- | -- | Dataflow | [100,101] |
| Apple | M1 | 10 | 5 | 119 | 64 | 11 | 1.1 | Dataflow | [102] |
| Apple | A14 | 6 | 5 | 88 | 64 | 11 | 1.83 | Dataflow | [103] |
| Apple | A15 | 7 | 5 | | 64 | 15.8 | 2.26 | Dataflow | [103] |
| Apple | A16 | 5.5 | 4 | | 64 | 17 | 3 | Dataflow | [104] |
| * AIStorm | AIStorm | 0.225 | | | 8 | 2.5 | 11 | Dataflow | [105] |
| * AlphaIC | RAP-E | 3 | | | 8 | 30 | 10 | Dataflow | [106] |
| aiCTX | Dynap-CNN | 0.001 | 22 | 12 | 1 | 0.0002 | 0.2 | Neuromorphic | [15,107] |
| * ARM | Ethos78 | 1 | 5 | | 16 | 10 | 10 | Dataflow | [108,109] |
| * AIMotive | Apache5 IEP | 0.8 | 16 | 121 | 8 | 1.6–32 | 2 | Dataflow | [110,111] |
| * Blaize | Pathfinder, EI Cano | 6 | 14 | | 64, FP-8, BF16 | 16 | 2.7 | Dataflow | [112] |
| *Bitman | BM1880 | 2.5 | 28 | 93.52 | 8 | 2 | 0.8 | Dataflow | [113,114] |
| * BrainChip | Akida1000 | 2 | 28 | 225 | 1, 2, 4 | 1.5 | 0.75 | Neuromorphic | [115,116] |
| * Cannan | Kendrite K210 | 2 | 28 | | 8 | 1.5 | 1.25 | Dataflow | [117,118] |
| * CEVA | CEVA-Neuro-S | | 16 | | 2, 5, 8, 12, 16 | 12.7 | | Dataflow | [119] |
| * CEVA | CEVA-Neuro-M | 0.83 | 16 | | 2, 5, 8, 12, 16 | 20 | 24 | Dataflow | [120] |
| * Cadence | DNA100 | 0.85 | 16 | | 16 | 4.6 | 3 | Dataflow | [121,122] |
| * Deepvision | ARA-1 | 1.7 | 28 | | 8, 16 | 4 | 2.35 | Dataflow | [123] |
| * Deepvision | ARA-2 | | 16 | | | | | Dataflow | [124] |
| * Eta | ECM3532 | 0.01 | 55 | 25 | 8 | 0.001 | 0.1 | Dataflow | [125] |
| * FlexLogic | InferX X1 | 13.5 | 7 | 54 | 8 | 7.65 | 0.57 | Dataflow | [126] |
| * Google | Edge TPU | 2 | 28 | 96 | 8, BF16 | 4 | 2 | Dataflow | [127,128] |
| * Gyrfalcon | LightSpeer 2803S | 0.7 | 28 | 81 | 8 | 16.8 | 24 | PIM | [47,89] |
| * Gyrfalcon | LightSpeer 5801 | 0.224 | 28 | 36 | 8 | 2.8 | 12.6 | PIM | [89] |
| * Gyrfalcon | Janux GS31 | 650/900 | 28 | 10457.5 | 8 | 2150 | 3.30 | PIM | [129] |
| * GreenWaves | GAP9 | 0.05 | 22 | 12.25 | 8, 16, 32 | 0.05 | 1 | Dataflow | [130–132] |
| * Horizon | Journey 3 | 2.5 | 16 | | 8 | 5 | 2 | Dataflow | [133] |
| * Horizon | Journey5/5P | 30 | 16 | | 8 | 128 | 4.8 | Dataflow | [134,135] |
| * Hailo | Hailo 8 M2 | 2.5 | 28 | 225 | 4, 8, 16 | 26 | 2.8 | Dataflow | [136,137] |
| Intel | Loihi 2 | 0.1 | 7 | 31 | 8 | 0.3 | 3 | Neuromorphic | [9] |
| Intel | Loihi | 0.11 | 14 | 60 | 1–9 | 0.03 | 0.3 | Neuromorphic | [9,138] |
| * Intel | Intel® Movidius | 2 | 16 | 71.928 | 16 | 4 | 2 | Dataflow | [139] |

| Company | Latest Chip | Max Power (W) | Process (nm) | Area (mm²) | Precision INT/FP | Max Performance (TOPS) | Energy Efficiency (TOPS/W) | Architecture | Reference |
|---|---|---|---|---|---|---|---|---|---|
| IBM | TrueNorth | 0.065 | 28 | 430 | 8 | 0.0581 | 0.4 | Neuroorphic | [10,138] |
| IBM | NorthPole | 74 | 12 | 800 | 2, 4, 8 | 200 (INT8) | 2.7 | Dataflow | [90,140] |
| * Imagination | PowerVR Series3NX | | | | FP-(8, 16) | 0.60 | | Dataflow | [141,142] |
| * Imec | DIANA | | 22 | 10.244 | 2 | 29.5 (A), 0.14 (D) | 14.4 | PIM + Digital | [143,144] |
| * Imagination | IMG 4NX MC1 | 0.417 | | | 4, 16 | 12.5 | 30 | Dataflow | [145] |
| * Kalray | MPPA3 | 15 | 16 | | 8, 16 | 255 | 1.67 | Dataflow | [13] |
| * Kneron | KL720 AI | 1.56 | 28 | 81 | 8, 16 | 1.4 | 0.9 | Dataflow | [47] |
| * Kneron | KL530 | 0.5 | | | 8 | 1 | 2 | Dataflow | [47] |
| * Koniku | Konicore | | | | | | | Neuromorphic | [12] |
| * LeapMind | Efficiera | 0.237 | 12 | 0.422 | 1, 2, 4, 8, 16, 32 | 6.55 | 27.7 | Dataflow | [21] |
| * Memryx | MX3 | 1 | -- | -- | 4, 8, 16, BF16 | 5 | 5 | Dataflow | [146] |
| * Mythic | M1108 | 4 | | 361 | 8 | 35 | 8.75 | PIM | [87] |
| * Mythic | M1076 | 3 | 40 | 294.5 | 8 | 25 | 8.34 | PIM | [18,86,88] |
| * mobileEye | EyeQ5 | 10 | 7 | 45 | 4, 8 | 24 | 2.4 | Dataflow | [147–149] |
| * mobileEye | EyeQ6 | 40 | 7 | | 4, 8 | 128 | 3.2 | Dataflow | [150] |
| * Mediatek | i350 | | 14 | | | 0.45 | | Dataflow | [151] |
| * NVIDIA | Jetson Nano B01 | 10 | 20 | 118 | FP16 | 1.88 | 0.188 | Dataflow | [152] |
| * NVIDIA | AGX Orin | 60 | 7 | -- | 8 | 275 | 3.33 | Dataflow | [153] |
| * NXP | i.MX 8M+ | | 14 | 196 | FP16 | 2.3 | | Dataflow | [84,85] |
| * NXP | i.MX9 | $4 \times 10^{-6}$ | 12 | | | | | Dataflow | [83] |
| * Perceive | Ergo | 0.073 | 5 | 49 | 8 | 4 | 55 | Dataflow | [154] |
| TSU & Polar Bear Tech | QM930 | 12 | 12 | 1089 | 4, 8, 16 | 20 (INT8) | 1.67 | Dataflow | [155] |
| Qualcomm | QCS8250 | | 7 | 157.48 | 8 | 15 | | Dataflow | [156,157] |
| Qualcomm | Snapdragon 888+ | 5 | 5 | | FP32 | 32 | 6.4 | Dataflow | [158–160] |
| Qualcomm | Snapdragon 8 Gen2 | | 4 | | 4, 8, 16, FP16 | 51 | | Dataflow | [161] |
| * RockChip | rk3399Pro | 3 | 28 | 729 | 8, 16 | 3 | 1 | Dataflow | [162] |
| Rokid | Amlogic A311D | | 12 | | | 5 | | Dataflow | [163] |
| Samsung | Exynos 2100 | | 5 | | | 26 | | Dataflow | [164,165] |
| Samsung | Exynos 2200 | | 4 | | 8, 16, FP16 | | | Dataflow | [166] |
| Samsung | HBM-PIM | 0.9 | 20 | 46.88 | | 1.2 | 1.34 | PIM | [167,168] |
| Sima.ai | MLSoC | 10 | 16 | 175.55 | 8 | 50 | 5 | Dataflow | [169,170] |
| Synopsis | EV7x | | 16 | | 8, 12, 16, | 2.7 | | Dataflow | [171,172] |
| * Syntiant | NDP100 | 0.00014 | 40 | 2.52 | | 0.000256 | 20 | PIM | [173,174] |
| * Syntiant | NDP101 | 0.0002 | 40 | 25 | 1, 2, 4,8 | 0.004 | 20 | PIM | [173,175] |
| * Syntiant | NDP102 | 0.0001 | 40 | 4.2921 | 1, 2, 4, 8 | 0.003 | 20 | PIM | [173,175] |
| * Syntiant | NDP120 | 0.0005 | 40 | 7.75 | 1, 2, 4, 8 | 0.0019 | 3.8 | PIM | [173,176] |

**Table 2.** *Cont.*

| Company | Latest Chip | Max Power (W) | Process (nm) | Area (mm²) | Precision INT/FP | Max Performance (TOPS) | Energy Efficiency (TOPS/W) | Architecture | Reference |
|---|---|---|---|---|---|---|---|---|---|
| * Syntiant | NDP200 | 0.001 | 40 | | 1, 2, 4, 8 | 0.0064 | 6.4 | PIM | [173,177] |
| Think Silicon | NEMA® | pico XS | 0.0003 | 28 | 0.11 | FP16, 32 | 0.0018 | 6 | Dataflow | [178] |
| Tesla/Samsung | FSD Chip | 36 | 14 | 260 | 8, FP-8 | 73.72 | 2.04 | Dataflow | [179] |
| Videntis | TEMPO | | | | | | | Neuromorphic | [11] |
| Verisilicon | VIP9000 | | 16 | | 16, FP16 | 0.5–100 | | Dataflow | [180,181] |
| Untether | TsunAImi | 400 | 16 | | 8 | 2008 | 8 | PIM | [182,183] |
| UPMEM | UPMEM-PIM | 700 | 20 | | 32, 64 | 0.149 | | PIM | [184–187] |

* Processors are available for purchase; Integer precision is indicated by only precision number(s). Floating point precision is denoted FP in the precision column.

Recent hardware studies show that lower precision does not have a major impact on inference accuracy. For example, Intel and Tsinghua University have presented QNAP [188], where they utilize 8 bits for weights and activations. They show an inference accuracy loss of only 0.11% and 0.40% for VGG-Net and GoogleNet, respectively, when compared to a software baseline with the ImageNet dataset. Samsung and Arizona State University have experimented with extremely-low-precision inference in PIMCA [189], where they utilized 1 bit for weights and activations. They showed that VGG-9 and ResNet-18 had accuracy losses of 3.89% and 6.02%, respectively.

Lower precision increases the energy and area efficiency of a system. PIMCA can compute 136 and 35 TOPS/W in 1- and 2-bit precision, respectively, for ResNet-18. TSMC [190] has studied the impact of low-precision computations on area efficiency. They showed 221 and 55 TOPS/mm² area efficiency in 4- and 8-bit precision. Thus, with 4-bit computation, they achieved about 3.5× higher computation throughput per unit area compared to 8-bit computation.

Brain-Float-16 (or BF-16) [191] is a limited precision floating point format that is becoming popular for AI applications in edge devices. BF16 combines certain components of FP32 and FP16. From FP16, the BF16 utilizes 16 bits overall. From FP32, BF16 utilizes 8 bits for the exponent field (instead of 5 bits for FP16). A key benefit of BF16 is the format obtains the same dynamic range and inference accuracy as FP32 [75]. BF16 speeds up the MAC operation in edge devices to enable faster AI inference on the edge devices. Both the GDDR6-AiM from SK Hynix [192] and Pathfinder-1600 from Blaize [112,193] support BF16 for AI applications. The supported precision levels of various edge processors are presented in Table 2.

### 3.2. Pruning

Pruning is the technique used to remove unnecessary network connections to make the network lightweight for deploying on edge processors. Several studies [92–100,194–196] show that up to 91% of weights in AlexNet can be pruned with minimal accuracy reduction. Various training methods have been proposed to apply pruning to pre-trained networks [99]. Pruning, however, has drawbacks such as creating sparsity in the weight matrices. This sparsity leads to unbalanced parallelism in the computation and irregular access to the on-chip memory. Several techniques have been developed [197,198] to reduce the sparsity.

### 3.3. Knowledge Distillation

Knowledge distillation, introduced by B. Christian et al. [199], is a technique wherein the knowledge of an ensemble of larger networks is transferred to a smaller network without loss of validity. This can reduce the computational load significantly. The effectiveness of knowledge distillation is studied extensively in [92–100,200–204], where the

authors show that the distillation of knowledge from a larger regularized model into a smaller model works effectively. Various algorithms have been proposed to improve the process of transferring knowledge, such as adversarial distillation, multi-teacher distillation, cross-modal distillation, attention-based distillation, quantized distillation, and NAS-based distillation [205]. Although knowledge distillation techniques are mainly used for classification applications, they are also applied to other applications, such as object detection, semantic segmentation, language modeling, and image synthesis [81].

## 4. Framework for Deep Learning Networks

At present, the majority of edge AI processors are designed for inference only. Network training is typically carried out on higher-performance desktop or server systems. There are a large variety of software frameworks used to train deep networks and also to convert them into lightweight versions suitable for edge devices. Popular DNN frameworks include Tensorflow (TF) [206], Tensorflow Lite (TFL) [207], PyTorch [208], PyTorch mobile [209], Keras [210], Caffe2 [211], OpenCV [212], ONNX [213], and MXNet [214]. Some of these frameworks support a broad class of devices, such as android, iOS, or Linux systems.

TFL was developed by Google and supports interfacing with many programming languages (such as Java, C++, and Python). It can take a trained model from TensorFlow and apply model compression to reduce the amount of computations needed for inference.

ONNX was developed by the PyTorch team to represent traditional machine learning and state-of-the-art deep learning models [213]. The framework is interoperable across popular development tools such as PyTorch, Caffe2, and Apache MXNet. Many of the current AI processors support the ONNX framework, such as Qualcomm SNPE, AMD, ARM, and Intel [215].

PyTorch mobile was developed by Facebook and allows a developer to train AI models for edge applications. The framework provides a node-to-node workflow that enables clients to have a privacy-preserving learning environment via collaborative or federated learning [208,209]. It supports XNNPACK floating point kernel libraries for ARM CPUs and integrates QNNPACK for quantized INT8 kernels [209].

Caffe2 is a lightweight framework developed by Facebook [211]. This framework supports C++ and Python APIs, which are interchangeable, and helps to develop prototypes quickly (such prototypes may potentially be optimized later). Caffe2 integrates with Android Studio and Microsoft Visual Studio for mobile development [211]. Caffe2Go is developed for embedding in mobile apps for applying a full-fledged deep learning framework for real-time capture, analysis, and decision making without the help of a remote server [216].

Facebook uses Pytorch Mobile, Caffe2 and ONNX for developing their products. Pytorch is used for experimental and rapid development, Caffe2 is developed for the production environment, and ONNX helps to share the models between the two frameworks [213].

MXNet is a fast and scalable framework developed by the Apache Software Foundation [131]. This framework supports both training and inference with a concise API for AI applications in edge devices. MXNet supports Python, R, C++, Julia, Perl, and many other languages and can be run on any processor platform for developing AI applications [131]. As shown in Table 3 TFL, ONNX, and Caffe2 are the most widely used frameworks for AI edge applications.

**Table 3.** Processors, supported neural network models, deep learning frameworks, and application domains.

| Company | Product | Supported Neural Networks | Supported Frameworks | Application/Benefits |
|---|---|---|---|---|
| Apple | Apple A14 | DNN | TFL | iPhone12 series |
| Apple | Apple A15 | DNN | TFL | iPhone13 series |

**Table 3.** *Cont.*

| Company | Product | Supported Neural Networks | Supported Frameworks | Application/Benefits |
|---|---|---|---|---|
| aiCTX-Synsense | Dynap-CNN | CNN, RNN, Reservoir Computing | SNN | High-speed aircraft, IoT, security, healthcare, mobile |
| ARM | Ethos78 | CNN and RNN | TF, TFL, Caffe2, PyTorch, MXNet, ONNX | Automotive |
| AIMotive | Apache5 IEP | GoogleNet, VGG16, 19, Inception-v4, v2, MobileNet v1, ResNet50, Yolo v2 | Caffe2 | Automotives, pedestrian detection, vehicle detection, lane detection, driver status monitoring |
| Blaize | EI Cano | CNN, YOLO v3 | TFL | Fit for industrial, retail, smart-city, and computer-vision systems |
| BrainChip | Akida1000 | CNN in SNN, MobileNet | MetaTF | Online learning, data analytics, security |
| BrainChip | AKD500, 1500, 2000 | DNN | MetaTF | Smart homes, smart health, smart city and smart transportation |
| CEVA | Neuro-s | CNN, RNN | TFL | IoTs, smartphones, surveillance, automotive, robotics, medical |
| Cadence | Tensilica DNA100 | FCC, CNN, LSTM | ONNX, Caffe2, TensorFlow | IoT, smartphones, AR/VR, smart surveillance, autonomous vehicles |
| Deepvision | ARA-1 | Deep Lab V3, Resnet-50, Resnet-152, MobileNet-SSD, YOLO V3, UNET | Caffe2, TFL, MXNET, PyTorch | Smart retail, robotics, industrial automation, smart cities, autonomous vehicles, and more |
| Deepvision | ARA-2 | Model in ARA-1 and LSTM, RNN, | TFL, Pytorch | Smart retail, robotics, industrial automation, smart cities, |
| Eta | ECM3532 | CNN, GRU, LSTM | --- | Smart homes, consumer products, medical, logistics, smart industry |
| Gyrfalcon | LightSpeer 2803S | CNN-based, VGG, ResNet, MobileNet; | TFL, Caffe2 | High-performance audio and video processing |
| Gyrfalcon | LightSpeer 5801 | CNN-based, ResNet, MobileNet and VGG16, | TFL, PyTorch & Caffe2 | Object detection and tracking, NLP, visual analysis |
| Gyrfalcon Edge Server | Janux GS31 | VGG, REsNet, MobileNet | TFL, Caffe2, PyTorch | Smart cities, surveillance, object detection, recognition |
| GreenWaves | GAP9 | CNN, LSTM, GRU, MobileNet | TF, Pytorch | DSP application |
| Horizon | Journey 3 | CNN, MobileNet v2, EfficientNet | TFL, Pytorch, ONNX, mxnet, Caffe2 | Automotives |
| Horizon | Journey5/5P | Resnet18, 50, MobileNet v1-v2, ShuffleNetv2, EfficientNet FasterRCNN, Yolov3 | TFL, Pytorch, ONNX, mxnet, Caffe2 | Automotives |

**Table 3.** *Cont.*

| Company | Product | Supported Neural Networks | Supported Frameworks | Application/Benefits |
|---------|---------|---------------------------|----------------------|----------------------|
| Hailo | Hailo 8 M2 | YOLO 3, YOLOv4, CenterPose, CenterNet, ResNet-50 | ONNX, TFL | Edge vision applications |
| Intel | Loihi 2 | SNN-based NN | Lava, TFL, Pytorch | Online learning, sensing, robotics, healthcare |
| Intel | Loihi | SNN-based NN | Nengo | Online learning, robotics, healthcare and many more |
| Imagination | PowerVR Series3NX | MobileNet v3, CNN | Caffe, TFL | Smartphones, smart cameras, drones, automotives, wearables |
| Imec & GF | DIANA | DNN | TFL, Pytorch | Analog computing in Edge inference |
| KoniKu | Konicore | Synthetic biology + silicon | -- | Chemical detection, aviation, security |
| Kalray | MPPA3 | Deep network converted to KaNN | Kalray's KANN | Autonomous vehicles, surveillance, robotics, industry, 5G |
| Kneron | KL720 AI | CNN, RNN, LSTM | ONNX, TFL, Keras, Caffe2 | Wide applications from automotive to home appliances |
| Kneron | KL520 | Vgg16, Resnet, GoogleNet, YOLO, Lenet, MobileNet, FCC | ONNX, TFL, Keras, Caffe2 | Automotives, homes, industry, and so on |
| LeapMind | Efficiera | CNN, YOLO v3, MobileNet v2, Lmnet | Blueoil, Python & C++ API | Homes, industrial machinery, surveillance cameras, robots |
| Memryx | MX3 | CNN | Pytorh, ONNX, TF, Keras | Automation, surveillance, agriculture, financial |
| Mythic | M1108 | CNN, large complex DNN, Resnet50, YOLO v3, Body25 | Pytorch, TFL, and ONNX | Machine vision, electronics, smart homes, UAV/drones, edge servers |
| Mythic | M1076 | CNN, complex DNN, Resnet50, YOLO v3 | Pytorch, TFL, and ONNX | Surveillance, vision, voice, smart homes, UAV, edge servers |
| MobileEye | EyeQ5 | DNN | | Autonomous driving |
| MobileEye | EyeQ6 | DNN | | Autonomous driving |
| Mediatek | i350 | DNN | TFL | Vision and voice, biotech and bio-metric measurements |
| NXP | i.MX 8M+ | DNN | TFL, Arm NN, ONNX | Edge vision |
| NXP | i.MX9 | CNN, MobileNet v1 | TFL, Arm NN, ONNX | Graphics, images, display, audio |
| NVIDIA | AGX Orin | DNN | TF, TFL, Caffe, Pytorch | Robotics, retail, traffic, manufacturing |
| Qualcomm | QCS8250 | CNN, GAN, RNN | TFL | Smartphones, tablets, supporting 5G, video and image processing |

**Table 3.** *Cont.*

| Company | Product | Supported Neural Networks | Supported Frameworks | Application/Benefits |
|---|---|---|---|---|
| Qualcomm | Snapdragon 888+ | DNN | TFL | Smartphones, tablets, 5G, gaming, video upscaling, image and video processing |
| RockChip | rk3399Pro | VGG16, ResNEt50, Inception4 | TFL, Caffe, mxnet, ONNX, darknet | Smart homes, cities, and industry, facial recognition, driving monitoring |
| Rokid | Amlogic A311D | Inception V3, YoloV2, YOLOV3 | TFL, Caffe2 Darknet | High-performance multimedia |
| Samsung | Exynos 2100 | CNN | TFL | Smartphones, tablets, advanced image signal processing (ISP), 5G |
| Samsung | HBM-PIM | DNN | Pytorch, TFL | Supercomputer and AI application |
| Synopsis | EV7x | CNN, RNN, LSTM | OpenCV, OpenVX and OpenCL C, TFL, Caffe2 | Robotics, autopilot cars, vision, SLAM, and DSP algorithms |
| Syntiant | NDP100 | DNN | TFL | Mobile phones, hearing equipment, smartwatches, IoT, remote controls |
| Syntiant | NDP101 | CNN, RNN, GRU, LSTM | TFL | Mobile phones, smart homes, remote controls, smartwatches, IoT |
| Syntiant | NDP102 | CNN, RNN, GRU, LSTM | TFL | Mobile phones, smart homes, remote controls, smartwatches, IoT |
| Syntiant | NDP120 | CNN, RNN, GRU, LSTM | TFL | Mobile phones, smart home, wearables, PC, IoT endpoints, media streamers, AR/VR |
| Syntiant | NDP200 | FC, Conv, DSConv, RNN-GRU, LSTM | TFL | Mobile phones, smart homes, security cameras, video doorbells |
| Think Silicon | Nema PicoXS | DNN | ---- | Wearable and embedded devices |
| Tesla | FSD | CNN | Pytorch | Automotives |
| Verisilicon | VIP9000 | All modern DNN | TF, Pytorch, TFL, DarkNet, ONNX | Can perform as intelligent eyes and intelligent ears at the edge |
| Untether | TsunAImi | DNN, ResNet-50, Yolo, Unet, RNN, BERT, TCNs, LSTMs | TFL, Pytorch | NLP, inference at the edge server or data center |
| UPMEM | UPMEM-PIM | DNN | ----- | Sequence alignment of DNA or protein, genome assembly, metagenomic analysis |

Some edge processors are compatible only with their in-home frameworks. For example, Kalray's MPPA3 edge processor is compatible with KaNN (Kalray Neural Network), so any trained deep network must be converted to KaNN to run on the MPPA3 processor [13].

CEVA introduced its own software framework, CEVA-DNN, for converting pre-trained network models and weights from offline training frameworks (such as Caffe, TensorFlow) for inference applications on the CEVA processors [119,217]. CEVA added a retrain feature in CEVA-DNN for the Neuro-Pro processor to enable a deployed device to be updated without uploading a database to the server [119]. The developer can also use CEVA-DNN tools on a simulator or test device and then transfer the updated model to edge devices [217].

## 5. Framework for Spiking Neural Networks

Spiking neural networks (SNNs) utilize brain-inspired computing primitives, where a neuron accumulates a potential and fires only when a threshold is crossed [218]. This means that in spiking neural networks, the neurons have outputs sporadically. Thus, SNNs have much fewer neuron-to-neuron communications compared to deep neural networks, where all neurons always send outputs. The net result of this is that SNNs can be dramatically more power-efficient than DNNs and can potentially implement a task with far fewer operations. Thus, an SNN processor with the same operations-per-second capability as a DNN processor could theoretically have a much higher task-level throughput.

To extract the highest efficiency from SNN processors, it is best to use algorithms that are developed from the ground up to use spiking neurons. Examples of such algorithms include constraint satisfaction problems [219] and genetic algorithms [220]. Several studies have examined how to implement DNNs using SNNs [221]. Davidson et al. [222] show through modeling of energies that this should not result in higher efficiency than the original DNN using the same underlying silicon technology. However, P. Blouw et al. [223] implemented keyword spotting on several hardware platforms and showed that the Loihi was about $5\times$ more energy-efficient than the Movidius deep learning processor. The remainder of this section describes some of the key frameworks for implementing SNN architectures for spiking neuromorphic processors.

Nengo is a Python-based framework developed by Applied Brain Research for spiking neurons. It supports multiple types of processors, including Loihi [224] and Spinnaker [225]. Nengo is very flexible in writing code and simulating SNNs. The core framework is the Nengo ecosystem, which includes Nengo objects and NumPy-based simulators. The Nengo framework has Nengo GUI for model construction and visualization tools and NengoDL for simulating deep learning models using SNNs [226].

Meta-TF [227] is a framework developed by BrainChip for edge applications in the Akida neuromorphic chips [115,116,228]. Meta-TF takes advantage of the Python scripting language and associated tools, such as Jupyter notebook and NumPy. Meta-TF includes three Python packages [227]: (1) the Akida Python package works as an interface to the Akida neuromorphic SoC; (2) the CNN2SNN tool provides an environment to convert a trained CNN network into SNNs. Brainchip embeds the on-chip training capability in the Akida processor, and thus, the developers can train SNNs on the Akida processor directly [228]; (3) Akida Model Zoo contains pre-created network models, which are built with the Akida sequential API and the CNN2SNN tool by using quantized Keras models.

Lava is a framework currently being developed by Intel to build SNN models and map them to neuromorphic platforms [229]. The current version of the Lava framework supports the Loihi neuromorphic chips [9]. Lava includes Magma which helps to map and execute neural network models and sequential processes to neuromorphic hardware [229]. Magma also helps to estimate performance and energy consumption on the platform. Lava has additional properties—including offline training, integration with other frameworks, and a Python interface—and is an open-source framework (with proper permissions). The Lava framework supports online real-time learning, where the framework adopts plasticity rules. However, the learning is constrained to access only locally available process information [229].

## 6. Edge Processors

At present, GPUs are the most popular platform for implementing DNNs. These, however, are usually not suitable for edge computing (except the NVIDIA Jetson systems) due to their high power consumption. A large variety of AI hardware has been developed, many of which target edge applications. Several articles have reviewed AI hardware in broad categories, giving an overall idea of the current trend in AI accelerators [230–232]. Earlier works [2,233–235] have reviewed a small selection of older edge AI processors.

This paper presents a very broad coverage of edge AI processors and PIM processors from the industry. This includes processors already released, processors that have been announced, and processors that have been published in research venues (such as the ISSCC and the VLSI conferences). The data presented here are collected from open-source platforms that include scientific articles, tech news portals, and company websites. The exact numbers could be different than in this report. If someone is interested in a particular processor, we suggest verifying the performance data with the providers. This section is divided into four subsections: subsection (i) describes dataflow processors; subsection (ii) describes neuromorphic processors; and subsection (iii) describes PIM processors. All of these sections describe industrial products that have been announced or released. Finally, subsection (iv) describes the processors in industrial research.

Table 2 describes the key hardware characteristics of the commercial edge-AI and PIM-AI processors. Table 3 lists the same key characteristics for the processors from industrial research. Table 4 describes the key software/application characteristics of the processors in Table 2.

**Table 4.** Edge processors in industrial research with technology, process technology, and numerical precision.

| Research Group | Name | Power (W) | Process (nm) | Area (mm$^2$) | Precision INT/FP * | Performance (TOPS) | E. Eff. (TOPS/W) | Architecture | Reference |
|---|---|---|---|---|---|---|---|---|---|
| TSMC + NTHU | | 0.00213 | 22 | 6 | 2, 4, 8 | 0.418 | 195.7 | PIM | [236] |
| TSMC | | 0.037 | 22 | 0.202 | 4, 8, 12, 16 | 3.3 | 89 | PIM | [237] |
| TSMC | | 0.00142 | 7 | 0.0032 | 4 | 0.372 | 351 | PIM | [238] |
| Samsung + GIT | FORMS | 66.36 | 32 | 89.15 | 8 | 0.0277 | | PIM | [239] |
| IBM + U Patra | HERMES | 0.0961 | 14 | 0.6351 | 8 | 2.1 | 21.9 | PIM | [240] |
| Samsung + ASU | PIMCA | 0.124 | | 20.9 | 1, 2 | 4.9 | 588 | PIM | [189] |
| Intel + Cornell U | CAPE | | 7 | 9 | 4 | | | PIM | [241] |
| SK Hynix | AiM | | | 6.08 | | 1 | | PIM | [192] |
| TSMC | DCIM | 0.0116 | 5 | 0.0133 | 4, 8 | 2.95 | 254 | PIM | [190] |
| Samsung | | 0.3181 | 4 | 4.74 | 4, 8, 16, FP16 | 39.3 | 11.59 | Dataflow | [242] |
| Alibaba + FU | | 0.0212 | 28 | 8.7 | 3 | 0.97 | 32.9 | Dataflow | [243] |
| Alibaba + FU | | 0.072 | 65 | 8.7 | 3 | 1 | 8.6 | Dataflow | [243] |
| Alibaba | | 0.978 | 55 | 602.22 | 8 | | | Dataflow | [244] |
| TSMC + NTHU | | 0.00227 | 22 | 18 | 2, 4, 8 | 0.91 | 960.2 | PIM | [245] |
| TSMC + NTHU | | 0.00543 | 40 | 18 | 2, 4, 8 | 3.9 | 718 | PIM | [246] |
| TSMC + GIT | | 0.000350 | 40 | 0.027 | | 0.0092 | 26.56 | PIM | [247] |
| TSMC + GIT | | 0.131 | 40 | 25 | 1–8, 1–8, 32 | 7.989 | 60.64 | PIM | [248] |
| Intel + UC | | 0.0090 | 28 | 0.033 | 1, 1 | 20 | 2219 | PIM | [249] |
| Intel + UC | | 0.0194 | 28 | 0.049 | 1–4, 1 | 4.8 | 248 | PIM | [249] |

**Table 4.** *Cont.*

| Research Group | Name | Power (W) | Process (nm) | Area (mm²) | Precision INT/FP * | Performance (TOPS) | E. Eff. (TOPS/W) | Architecture | Reference |
|---|---|---|---|---|---|---|---|---|---|
| TSMC + NTHU | nvCIM | 0.00398 | 22 | 6 | 2,4 | 5.12 | 1286.4 | PIM | [69] |
| Pi2star + NTHU | | 0.00841 | 65 | 12 | 1–8 | 3.16 | 75.9 | PIM | [250] |
| Pi2star + NTHU | | 0.00652 | 65 | 9 | 4, 8 | 2 | 35.8 | PIM | [251] |
| Tsing + NTHU | | 0.273 | 28 | 6.82 | 12 | 4.07 | 27.5 | Dataflow | [252] |
| Samsung | | 0.381 | 4 | 4.74 | 4, 8, FP16 | 19.7 | 11.59 | Dataflow | [242] |
| Renesas Electronics | | 4.4 | 12 | | | 60.4 | 13.8 | Dataflow | [253] |
| IBM | | 6.20 | 7 | 19.6 | 2, 4, FP(8,16,32) | 102.4 | 16.5 | Dataflow | [254] |
| Intel + IMTU | QNAP | 0.132 | 28 | 3.24 | 8 | 2.3 | 17.5 | Dataflow | [188] |
| Samsung | | 0.794 | 5 | 5.46 | 8, 16 | 29.4 | 13.6 | Dataflow | [255] |
| Sony | | 0.379 | 22 | 61.91 | 8, 16, 32 | 1.21 | 4.97 | Dataflow | [256] |
| Mediatek | | 1.05 | 7 | 3.04 | | 3.6 | 13.32 | Dataflow | [257] |
| Pi2star | | 0.099 | 65 | 12 | 8 | 1.32 | 13.3 | Dataflow | [74] |
| Mediatek | | 0.0012 | 12 | | | 0.102 | 86.24 | PIM | [257] |
| TSMC + NTHU | | 0.10 | 22 | 8.6 | 8, 8, 8 | 6.96 | 68.9 | PIM | [258] |
| TSMC + NTHU | | 0.099 | 22 | 9.32 | 8, 8, 8 | 24.8 | 251 | PIM | [258] |
| ARM + Harvard | | 0.04 | 12 | | FP4 | 0.734 | 18.1 | Dataflow | [259] |
| ARM + Harvard | | 0.045 | 12 | | FP8 | 0.367 | 8.24 | Dataflow | [259] |
| TSMC + NTHU | | 0.0037 | 22 | 18 | 8, 8, 22 | 0.59 | 160.1 | Dataflow | [260] |
| STMircroelectronics | | 0.738 | 18 | 4.24 | 1, 1 | 229 | 310 | Dataflow | [261] |
| STMircroelectronics | | 0.740 | 18 | 4.19 | 4, 4 | 57 | 77 | Dataflow | [261] |
| MediaTek | | 0.711 | 12 | 1.37 | 12 | 16.5 | 23.2 | PIM | [262] |
| TSMC+ NTHU | | | 16 | | 8 | | 98.5 | PIM | [263] |
| Renesas Electronics | | 5.06 | 14 | | 8 | 130.55 | 23.9 | Dataflow | [264] |

* Integer precision is indicated by only precision number(s). Floating point precision is denoted FP in the precision column.

## 6.1. Dataflow Edge Processor

This section describes the latest dataflow processors from the industry. Dataflow processors are custom-designed for neural network inference and, in some cases, training computations. The processors are listed in alphabetical order based on the manufacturer name. The data provided are from the publications or websites of the processors.

Analog Devices Inc. developed a low-cost mixed-signal CNN accelerator MAX78000 that consists of a Cortex-M4 processor, a 32-bit RISC-V processor with a floating point unit (FPU) and CPU for system control with a DNN accelerator [100,101]. The accelerator has a SRAM-based 442 KB on-chip weight storage memory which can support 1-, 2-, 4-, and 8-bit weights. The CNN engine has 64 parallel processors and 512 KB data memory. Each processor has a pooling unit and a convolutional unit with a dedicated memory unit. The processor consumes 1 pJ/MAC operation. As the exact power consumption (W) and performance (TOPS) data are not publicly available at the time of this writing, we did not add it to our graphs. The size of the chip is 64 mm². The architecture supports Pytorch and Tensorflow toolsets for the development of a range of DNN models. The target application

areas are object detection, classification, facial recognition, time series data processing, and noise cancellation.

Apple released the bionic SoC A16 with an NPU unit for the iPhone 14 [104]. The A16 processor exhibits about 20% better performance with the same power consumption as their previous version, A15. It is embedded with a 6-core ARM8.6a CPU, 16-core NPU, and 8-core GPU [104]. The Apple M2 processor was released in 2022 primarily for Macbooks and then optimized for iPads. This processor includes a 10-core GPU and 16-core NPU [265]. M1 performs 11 TOPS with 10 W of power consumption [109]. The M2 has an 18% and 35% more powerful CPU and GPU for faster computation.

ARM recently announced the Ethos-N78 with an 8-core NPU for automotive applications [108]. Ethos-N78 is an upgraded version of Ethos-N77. Both NPUs support INT8 and INT16 precision. Ethos-N78 performs more than two times better than the earlier version. The most significant improvement of Ethos-N78 is a new data compression method that reduces the bandwidth and improves performance and energy efficiency [109].

Blaize released its Pathfinder P1600 El Cano AI inference processor. This processor integrates 16 graph streaming processors (GSPs) that deliver 16 TOPS at its peak performance [112]. It uses a dual Cortex-A53 for running the operating system at up to 1 GHz. Blaize GSP processors integrate data pipelining and support up to INT-64 and FP-8-bit operations [112].

AIMotive [110] introduced the inference edge processor Apache5, which supports a wide range of DNN models. The system has an aiWare3p NPU with an energy efficiency of 2 TOPS/W. Apache5 supports INT8 MAC and INT32 internal precision [111]. This processor is mainly targeted at autonomous vehicles [266].

CEVA [119] released the Neupro-S on-device AI processor for computer vision applications. Neupro comprises two separate cores. One is the DSP-based Vector Processor Unit (VPU), and the other is the Neupro Engine. VPU is the controller, and the Neupro Engine performs most of the computing work with INT8 or INT16 precision. A single processor performs up to 12.5 TOPS, while the performance can be scaled to 100 TOPS with multicore clusters [119,120]. The deep learning edge processors are mostly employed for inference tasks. CEVA added a retraining capability to its CDNN (CEVA DNN) framework for online learning on client devices [217].

Cadence introduced the Tensilica DNA 100, which is a comprehensive SoC for domain-specific on-device AI edge accelerators [121]. It has low-, mid-, and high-end AI products. Tensilica DNA 100 offers 8 GOPS to 32 TOPS AI processing performance currently and predicts 100 TOPS in future releases [122]. The target applications of the DNA 100 include IoTs, intelligent sensors, vision, and voice application. The mid- and high-end applications include smart surveillance and autonomous vehicles, respectively.

Deepvision has updated their edge inference coprocessor ARA-1 for applications to autonomous vehicles and smart industries [123]. It includes eight compute engines with 4 TOPS and consumes 1.7–2.3 W of power [123]. The computing engine supports INT8 and INT16 precision. Deepvision has recently announced its second-generation inference engine, ARA-2, which will be released later in 2022 [124]. The newer version will support LSTM and RNN neural networks in addition to the networks supported in ARA-1.

Horizon announced its next automotive AI inference processor Journey 5/5P [133], which is the updated version of Journey 3. The mass production of Journey 5 will be starting in 2022. The processor exhibits a performance of 128 TOPS, and has a power of 30 W, giving an energy efficiency of 4.3 TOPS/W [134,135].

Hailo released its Hailo-8 M-2 SoC for various edge applications [136]. The computing engine supports INT8 and INT16 precision. This inference engine is capable of 26 TOPS and requires 2.5 W of power. The processor can be employed as a standalone or coprocessor [137].

Google introduced its Coral Edge TPU, which comprises only 29% of the floorplan area of the original TPU for edge applications [127]. The Coral TPU shows high energy efficiency in DNN computations compared to the original TPUs which are used in cloud

inference applications [267]. Coral Edge TPU supports INT8 precision and can perform 4 TOPS with 2 Watts of power consumption [127].

Google released its Tensor processor for mobile applications, coming with its recent Pixel series mobile phone [268]. Tensor is an 8-core cortex CPU chipset fabricated with 5 nm process technology. The processor has a 20-core Mali-G78 MP20 GPU with 2170 GFLOPS computing speed. The processor has a built-in NPU to accelerate AI models with a performance of 5.7 TOPS. The maximum power consumption of the processor is 10 W.

GreenWaves announced their edge inference chip GAP9 [130]. It is a very low-cost, low-power device that consumes 50 mW and performs 50 GOPS at its peak [132]. However, it consumes 330μW/GOP [131]. GAP9 provides hearable developments through DSP, AI accelerator, and ultra-low-latency audio streaming on IoT devices. GAP9 supports a wide range of computing precision, such as INT8, 16, 24, 32, and FP16, 32 [131].

IBM introduced the NorthPole [90,140], a non-von Neumann deep learning inference engine, at the HotChips 2023 conference. The processor shows massive parallelism with 256 cores. Each core has 768 KB of near-computer memory to store weights, activations, and programs. The total on-chip memory capacity is 192 MB. The NorthPole processor does not use off-chip memory to load weights or store intermediate values during deep learning computations. Thus, it dramatically improves latency, throughput, and energy consumption, which helps outperform existing commercial deep learning processors. The external host processor works on three commands: write tensor, run network, and read tensor. The NorthPole processor follows a set of pre-scheduled deterministic operations in the core array. It is implemented in 12 nm technology and has 22 billion transistors taking up 800 mm$^2$ of chip area. The performance data released on the NothPole processor are computed based on frame/sec. The performance metrics of operations/sec in integer or floating point are unavailable in the public domain currently. However, the operations per cycle are available for different data precisions. In vector–matrix multiplication, 8-, 4-, and 2-bit cases can perform 2048, 4096, and 8192 operations/cycle. The FP16 can compute 256 operations/cycle (the number of cycles/s has not been released at this time). NorthPole can compute 800, 400, and 200 TOPS with INT 2, 4, and 8 precisions. The processor can be applied to a broad area of applications and can execute inference with a wide range of network models applied in classification, detection, segmentation, speech recognition, and transformer models in NLP.

Imagination introduced a wide range of edge processors with targeted applications in IoTs to autonomous vehicles [182]. The edge processor series is categorized as the PowerVR Series3NX and can achieve up to 160 TOPS with multicore implementations. For ultra-low-power applications, one can choose PowerVR AX3125, which has a 0.6 TOPS computing performance [183]. IMG 4NX MC1 is a single-core Series 4 processor for autonomous vehicle applications and performs at 12.5 TOPS with less than 0.5 W of power consumption [184].

Intel released multiple edge AI processors such as Nirvana Spring Crest NNP-I [269] and Movidious [139]. Recently, they have announced a scalable fourth-generation Xeon processor series that can be used for desktop to extreme edge devices [270]. The power consumption for an ultra-mobile processor is around 9 W when computed with INT8 precision. The development utilizes the SuperFin fabrication technology with 10 nm process technology. Intel is comparing its core architecture to the Skylake processor, and it claims an efficient core achieves 40% better performance with 40% less power.

IBM developed the Artificial Intelligence Unit (AIU) based on their AI accelerator used in the 7-nanometer Telum chip that powers its z16 system [271]. AIU is a scaled version developed using 5 nm process technology and features a 32-core design with a total of 23 billion transistors. AIU uses IBM's approximate computing frameworks where the computing executes with FP16 and FP32 precisions [272].

Leapmind has introduced the Efficiera for edge AI inference implemented in FPGA or ASIC [21]. Efficiency is for ultra-low-power applications. The computations are typically performed in 8-, 16-, or 32-bit precision. However, the company claims that 1-bit weight and 2-bit activation can be achieved while still maintaining accuracy for better power and

area efficiency. They show 6.55 TOPS at 800 MHz clock frequency with an energy efficiency of 27.7 TOPS/W [273].

Kneron released its edge inference processor, KL 720, for various applications, such as autonomous vehicles and smart industry [47]. The KL 720 is an upgraded version of the earlier KL 520 for similar applications. The revised version performs at 0.9 TOPS/W and shows up to 1.4 TOPS. The neural computation supports INT8 and INT16 precisions [47]. Kneron's most up-to-date heterogeneous AI chip is KL 530 [47]. It is enabled with a brand new NPU, which supports INT4 precision and offers 70% higher performance than that of INT8. The maximum power consumption of KL 530 is 500 mW and can deliver up to 1 TOPS [47].

Memryx [146] released an inference processor, MX3. This processor computes deep learning models with 4-, 8-, or 16-bit weight and BF16 activation functions. MX3 consumes about 1 W of power and computes with 5 TFLOPS. This chip stores 10 million parameters on a die and thus needs more chips for implementing larger networks.

MobileEye and STMicroelectronics released EyeQ 5 SoC for autonomous driving [147]. EyeQ 5 is four times faster than their earlier version, EyeQ 4. It can produce 2.4 TOPS/W and goes up to 24 TOPS with 10 W of power [148]. Recently, MobileEye has announced their next-generation processor, EyeQ6, which is around $5\times$ faster than EyeQ5 [149]. For INT8 precision, EyeQ5 performs 16 TOPS, and EyeQ6 shows 34 TOPS [150].

NXP introduced their edge processor i.MX 8M+ for targeted applications in vision, multimedia, and industrial automations [84]. The system includes a powerful Cortex-A53 processor integrated with an NPU. The neural network performs 2.3 TOPS with 2 W of power consumption. The neural computation supports INT16 precision [85]. NXP is scheduled to launch its next AI processor, iMX9, in 2023, with more features and efficiency [84].

NVIDIA released the Jetson Nano, which can run multiple applications in parallel, such as image classification, object detection, segmentation, and speech processing [152]. This developer kit is supported by the NVIDIA JetPack SDK and can run state-of-the-art AI models. The Jetson Nano consumes around 5–10 W of power and computes 472 GFLOPS in FP16 precision. The new version of Jetson Nano B01 can perform 1.88 TOPS [274].

NVIDIA released Jetson Orin, which includes specialized development hardware, AGX Orin. It is embedded with 32 GB of memory, has a 12-core CPU, and can exhibit a computing performance of 275 TOPS while using INT8 precision [152]. The computer is powered by NVIDIA ampere architecture with 2048 cores, 64 tensor cores, and 2 NVDLA v2.0 accelerators for deep learning [153].

Qualcomm developed the QCS8250 SoC for intensive camera and edge applications [156]. This processor supports Wi-Fi and 5G for the IoTs. A quad hexagon vector extension V66Q with hexagon DSP is used for machine learning. An integrated NPU is used for advanced video analysis. The NPU supports INT8 precision and runs at 15 TOPS [157].

Qualcomm has released the Snapdragon 888+ 5G processor for use in smartphones. It takes the smartphone experience to a new level with AI-enhanced gaming, streaming, and photography [158]. It includes a sixth-generation Qualcomm AI engine with the Qualcomm Hexagon780 CPU [159,160]. The throughput of the AI engine is 32 TOPS with 5 W of power consumption [159]. The Snapdragon 8 Gen2 mobile platform was presented at the HotChips 2023 conference and exhibited 60% better energy efficiency than the Snapdragon 8 in INT4 precision.

Samsung announced the Exynos 2100 AI edge processor for smartphones, smartwatches, and automobiles [164]. Exynos supports 5G network and performs on-device AI computations with triple NPUs. They fabricate using 5 nm extreme UV technology. The Exynos 2100 consumes 20% lower power and delivers 10% higher performance than Exynos 990. Exynos 2100 can perform up to 26 TOPS, and it is two times more power-efficient than the earlier version of Exynos [165]. A more powerful mobile processor, Exynos 2200, was released recently.

SiMa.ai [169] introduced the MLSoC for computer vision applications. MLSoc is implemented on TSMC 16 nm technology. The accelerator can compute 50 TOPS while

consuming 10 W of power. MLSoC uses INT8 precision in computation. The processor has 4 MB of on-chip memory for deep learning operations. The processor is 1.4× more efficient than Orin, measured in frames/W.

Tsinghua and Polar Bear Tech released their QM930 accelerator consisting of seven chiplets [155]. The chiplets are organized as one hub chiplet and six side chiplets, forming a hub-side processor. The processor is implemented in 12 nm CMOS technology. The total area for the chiplets is 209 mm$^2$ for seven chiplets. However, the total substrate area of the processor is 1089 mm$^2$. The processor can compute with INT4, INT8, and INT16 precision, showing peak performances of 40, 20, and 10 TOPS, respectively. The system energy efficiency is 1.67 TOPS/W when computed in INT8. The power consumption can be varied from 4.5 to 12 W.

Verisilicon introduced VIP 9000 for face and voice recognition. It adopts Vivante's latest VIP V8 NPU architecture for processing neural networks [180]. The computing engine supports INT8, INT16, FP16, and BF16. The performance can be scaled from 0.5 to 100 TOPS [181].

Synopsis developed the EV7x multi-core processor family for vision applications [171]. The processor integrates vector DSP, vector FPU, and a neural network accelerator. Each VPU supports a 32-bit scalar unit. The MAC can be configured for INT8, INT16, or INT32 precisions. The chip can achieve up to 2.7 TOPS in performance [172].

Tesla designed the FSD processor which was manufactured by Samsung for autonomous vehicle operations [179]. The SoC processor includes two NPUs and one GPU. The NPUs support INT8 precision, and each NPU can compute 36.86 TOPS. The peak performance of the FSD chip is 73.7 TOPS. The total TDP power consumption of each FSD chip is 36 W [179].

Several other companies have also developed edge processors for various applications but did not share hardware performance details on their websites or through publicly available publications. For instance, Ambarella [275] has developed various edge processors for automotive, security, consumer, and IoTs for industrial and robotics applications. Ambarella's processors are SoC types, mainly using ARM processors and GPUs for DNN computations.

### 6.2. Neuromorphic Edge AI Processor

In 2022, the global market value of neuromorphic chips was USD 3.7 billion, and by 2028, the estimated market value is projected to be USD 27.85 billion [276]. The neuromorphic processors described in this section utilize spike-based processing.

Synsense (formerly AICTx) has introduced a line of ultra-low-power neuromorphic processors: DYNAP-CNN, XYLO, DYNAP-SE2, and DYNAP-SEL [15]. Of these, we were able to find performance information on only the DYNAP-CNN chip. This processor is fabricated on a 22 nm process technology and has a die area of 12 mm$^2$. Each chip can implement up to a million spiking neurons, and a collection of DYNAP-CNN chips can be utilized to implement a larger CNN architecture. The chip utilizes asynchronous processing circuits [107].

BrainChip introduced the Akida line of spiking processors. The AKD1000 has 80 NPUs, 3 pJ/synaptic operation, and around 2 W of power consumption [115]. Each NPU consists of eight neural processing engines that run simultaneously and control convolution, pooling, and activation (ReLu) operations [116]. Convolution is normally carried out in INT8 precision, but it can be programmed for INT 1, 2, 3 or 4 precisions while sacrificing 1–3% accuracy. BrainChip has announced future releases of smaller and larger Akida processors under the AKD500, AKD1500, and AKD2000 labels [116]. A trained DNN network can be converted to SNN by using the CNN2SNN tool in the Meta-TF framework for loading a model into an Akida processor. This processor also has on-chip training capability, thus allowing the training of SNNs from scratch by using the Meta-TF framework [227].

GrAI Matters Lab (GML) developed and optimized a neuromorphic SoC processor named VIP for computer vision application. VIP is a low-power and low-latency AI

processor with 5–10 W of power consumption, and its latency is 10× less than the NVIDIA nano [277]. The target applications are for audio/video processing on end devices.

IBM developed the TrueNorth neuromorphic spiking system for real-time tracking, identification, and detection [10]. It consists of 4096 neurosynaptic cores and 1 million digital neurons. The typical power consumption is 65 mW, and the processor can execute 46 GSOPS/W, with 26 pJ per synaptic operation [10,278]. The total area of the chip is 430 mm$^2$, which is almost 14× bigger than that of Intel's Loihi 2.

Innatera announced a neuromorphic chip that is fabricated using TSMC's 28 nm process [279]. When tested with audio signals [280], each spike event consumed about 200 fJ, while the chip consumed only 100 uW for each inference event. The target application areas are mainly audio, healthcare, and radar voice recognition [280].

Intel released the Loihi [9], a spiking neural network chip, in 2018 and an updated version, the Loihi 2 [9], in 2021. The Loihi 2 is fabricated using Intel's 7 nm technology and has 2.3 billion transistors with a chip area of 31 mm$^2$. This processor has 128 neuron cores and 6 low-power x86 cores. It can evaluate up to 1 million neurons and 120 million synapses. The Loihi chips support online learning. Loihi processors support INT8 precision. Loihi 1 can deliver 30 GSOPS with 15 pJ per synaptic operation [138]. Both Loihi 1 and Loihi 2 consume similar amounts of power (110 mW and 100 mW, respectively [221]). However, the Loihi 2 outperforms the Loihi 1 by 10 times. The chips can be programmed through several frameworks, including, Nengo, NxSDK, and Lava [229]. The latter is a framework developed by Intel and is being pushed as the primary platform to program the Loihi 2.

IMEC developed a RISC-V processor-based digital neuromorphic processor with 22 nm process technology in 2022 [281]. They implemented an optimized BF-16 processing pipeline inside the neural process engine. The computation can also support INT4 and INT8 precision. They used three-layer memory to reduce the chip area.

Koniku combines biological machines with silicon devices to design a micro electrode array system core [12]. They are developing hardware and an algorithm that mimic the smell sensory receptor that is found in some animal noses. However, the detailed device parameters are not publicly available. The device is mainly used in security, agriculture, and safe flight operation [282].

*6.3. PIM Processor*

PIM processors are becoming an alternative for AI application due to their low latency, high energy efficiency, and reduced memory requirements. PIMs are in-place computing architectures that can be analog, and they reduce the burden of additional storage modules. However, there are some digital PIM systems have been developed, and schematic representations of a common PIM computing architectures have been presented. These systems consist of a crossbar array (N × M) of any popular storage devices. The crossbar array performs as the weight storage and analog multiplier. The storage devices could be SRAM, RRAM, PCM, STT-MRAM or a flash memory cell. The computing array is equipped with the peripheral circuits, a data converter (ADC or DAC), sensing circuits, and written circuits for the crossbar. Some of the PIM processors are discussed in this section.

Imec and GlobalFoundries have developed DIANA, a processor that includes both digital and analog cores for DNN processing. The digital core is employed for widely parallel computation, whereas the analog in-memory computing (AiMC) core enables much higher energy efficiency and throughput. The core uses a 6T-SRAM array with a size of 1152 × 512. Imec developed the architecture, while the chip is fabricated using GlobalFoundries' 22FDX solution [143]. It is targeted at a wide range of edge applications, from smart speakers to self-driving vehicles. The analog component (AiMC) computes at 29.5 TOPS with and the digital core computes at 0.14 TOPS. The digital and analog components have efficiencies of 4.1 TOPS/W and 410 TOPS/W, respectively in isolation. The overall system energy efficiency of DIANA is 14.4 TOPS/W for Cifar-10 [144].

Gyrfalcon has developed multiple PIM processors, including the Lightspeeur 5801, 2801, 2802, and 2803 [24]. The architecture uses digital AI processing in-memory units that

compute a series of matrices for CNN. The Lightspeeur 5801 has a performance of 2.8 TOPS at 224 mW and can be scaled up to 12.6 TOPS/W. The Lightspeeur 2803S is their latest PIM processor for the advanced edge, desktop, and data center deployments [19]. Each Lightspeeur 2803S chip performs 16.8 TOPS while consuming 0.7 W of power, giving an efficiency of 24 TOPS/W. Lightspeeur 2801 can compute 5.6 TOPS with an energy efficiency of 9.3 TOPS/W. Gyrfalcon introduced its latest processor, Lightspeeur 2802, using TSMC's magnetoresistive random access memory technology. Lightspeeur 2802 exhibits an energy efficiency of 9.9 TOPS/W. Janux GS31 is the edge inference server which is built with 128 Lightspeeur 2803S chips [129]. It can perform 2150 TOPS and consumes 650 W.

Mythic has announced its new analog matrix processor, M1076 [18]. The latest version of Mythic's PIM processor reduced its size by combining 76 analog computing tiles, while the original one (M1108) uses 108 tiles. The smaller size offers more compatibility to implant on edge devices. The processor supports 79.69 M on-chip weights in the array of flash memory and 19,456 ADCs for parallel processing. There is no external DRAM storage required. The DNN models are quantized from FP32 to INT8 and retrained in Mythic's analog compute engine. A single M1076 chip can deliver up to 25 TOPS while consuming 3 W of power [88]. The system can be scaled for high performance up to 400 TOPS by combining 16 M1076 chips which require 75 W [86,87].

Samsung has announced its HBM-PIM machine learning-enabled memory system with PIM architecture [16]. This is the first successful integration of a PIM architecture of high bandwidth memory. This technology incorporates the AI processing function into the Samsung HBM2 Aquabolt to speed up high-speed data processing in supercomputers. The system delivered 2.5× better performance with 60% lower energy consumption than the earlier HBM1 [16]. Samsung LPDDR5-PIM memory technology for mobile device technology is targeted at bringing AI capability into mobile devices without connecting to the data center [167]. The HBM-PIM architecture is different from the traditional analog PIM architecture, as outlined in Figure 2. It does not require data conversion and sensing circuits as the actual computation takes place in the near-computing module in the digital domain. Instead, it uses a GPU surrounded by HBM stacks to realize the parallel processing and minimize data movement [168]. Therefore, this is similar to a dataflow processor.

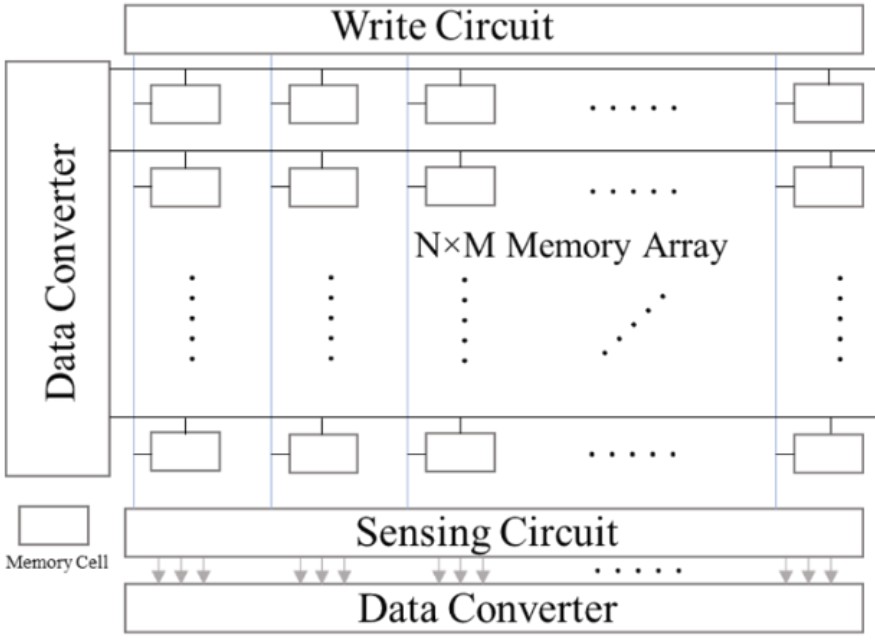

**Figure 2.** Schematic representation of a processing-in-memory macro system.

Syntiant has developed a line of flash memory array-based edge inference processors, such as NDP10x, NDP120, NDP200 [173]. Syntiant's PIM architecture is very energy-efficient and it combines with an edge-optimized training pipeline. A Cortex-M0 is embedded in the system that runs the NDP firmware. The NDP10x processors can hold 560 k weights of INT4 precision and perform MAC operation with an INT8 activation. The training pipeline can build neural networks for various applications according to the specifications with optimized latency, memory size, and power consumption [173]. Syntiant released five different versions of application processors. NDP 100 is their first AI processor, updated in 2020 with a tiny dimension of 2.52 mm$^2$ and ultra-low power consumption, less than 140 μW [174]. Syntiant continues to provide more PIM processors named NDP 101, 102, 120, and NDP 200 [175,177,283]. The application domains are mainly smartphones, wearable and hearable pieces of equipment, remote controls, and IoT endpoints. The neural computations are supported by INT 1, 2, 4, and 8 precision. The energy efficiency of the NDP 10× series is 2 TOPS/W [284], which includes NDP100, NDP 101, and NDP 102. NDP 120 [175] and NDP 200 exhibit 1.9 GOPS/W and 6.4 GOPS/W [177], respectively.

Untether has developed its PIM AI accelerator card TsunAImi [182] for inference at the data center or in the server. The heart of the TsunAImi is four runAI200 chips which are fabricated by TSMC in standard SRAM arrays. Each runAI200 chip features 511 cores and 192 MB of SRAM memory. runAI200 computes in INT8 precision and performs 502 TOPS at 8 TOPS/W, which is 3× more than NVIDIA's Ampere A100 GPU. The resulting performance of TsunAImi system is 2008 TOPS with 400 W [183].

UPMEMP PIM innovatively placed thousands of DPU units within the DRAM memory chips [184]. The DPUs are controlled by high-level applications running on the main CPU. Each DIMM consists of 16 PIM-enabled chips. Each PIM has 8 DPUs; thus, 128 DPUs are contained in each UPMEM [185].

However, the system is massively parallel, and up to 2560 DPUs units can be assembled as a unit server with 256 GB PIM DRAM. The computing power is 15× of a x86 server with the main CPU. The throughput benchmarked for INT32 bit addition is 58.3 MOPS/DPU [186]. This system is suitable for DNA sequencing, genome comparison, phylogenetics, metagenomic analysis, and more [187].

*6.4. Processors in Industrial Research*

The PIM computing paradigm is still in its rudimentary stage; however, it is a very promising system for efficient MAC operation and low-power edge application. A good number of industries and industry–academic research collaborations are escalating the development of PIM technologies and architectures. In this section, PIM processors in industry and industry–university collaboration are briefly discussed. The recent developments of PIM research are tabulated in Table 4.

Alibaba has developed SRAM and DRAM-based digital CIM and PNM systems for low-precision edge applications [243]. The CIM architecture uses multiple chiplet modules (MCMs) to solve the complex problem instead of a single SoC. The CIM architecture in [247] proposes a computing-on-memory boundary (COMB), which is a compromise between in-memory and near-memory computation. This technique exhibits high macro computing energy efficiency and low system power overhead. This CIM architecture demonstrated scalable MCM systems using a COMB NN processor. Layerwise pipeline mapping schemes are utilized to deploy different sizes of NNs for the required operation. The chip operation is demonstrated with keyword spotting, CIFAR-10 image classification, and object detection with tiny-YOLO NN using one, two, and four chiplets.

IBM and the University of Patra together presented their PCM-based CIM processor, HERMES [240]. This CIM is a 256 × 256 in-memory computed core fabricated in a 14 nm CMOS process technology for edge inference. HERMES is demonstrated for image classification operation on MNIST and CIFAR-10 datasets.

Samsung technology has been working on various CIM architectures for AI applications for the edge to the datacenter. The company has released HBM-PIM recently [167,168].

HBM-PIM is for high-speed memory access, which is fabricated with DRAM in a 20 nm process. Samsung and Arizona State University (ASU) presented a PIMCA chip for AI inference [189]. PIMCA consumes a very low amount of power (124 mW). PIMCA is highly energy-efficient (588 TOPS/W), as shown in Table 2. TSMC has designed and fabricated analog [237,238] and digital [190] CIM systems for inference.

Besides TSMC's own research, the company has multiple CIM research projects on various emerging memory devices such as ReRAM [235], STT-MRAM [245], PCM [246], RRAM [247], and RRAM-SRAM [248] in collaboration with various research groups in the academia. The performance of these macro inference chips has been demonstrated in various high-tier conferences or scientific forums very recently. The best performance was demonstrated in ISSCC 2022 with PCM devices, and it exhibited 5.12 TOPS in 2-bit precision [69], which was 1286.4 TOPS/W. This CIM processor supports INT2 and 4-bit computing precision. The digital CIM system is fabricated with FinFETs in 5 nm process technology, and it performs at 2.95 TOPS and 254 TOPS/W [190].

In addition to the AI accelerators introduced above, there are a handful of companies that are working on edge processors. The companies working on neuromorphic processors are MemComputing [107,285], GrAI [277], and iniLabs [286]. Memryx is a recently formed a startup which is building high-performance and energy-efficient AI processors for a wide range of applications, such as transportation, IoT, and industry [149]. It can compute Bfloat16 activation with 4/8/16-bit weight and performs at about 5TFLOPS.

## 7. Performance Analysis of Edge Processors

This section discusses the performance analysis of the edge processors described earlier. The discussion is focused on different architectures for edge processors. At first, overall performance is discussed based on the computing performance, power consumption, chip area, and computing precision. Then, only PIM processors are discussed. At the end of this section, we focus on the devices still under research and development or awaiting commercial availability.

### 7.1. Overall Analysis of AI Edge Processors

We compare all the edge AI processors listed in the previous section using the following key metrics:

1.  Performance: tera-operations per second (TOPS);
2.  Energy efficiency: TOPS/W;
3.  Power: Watt (W);
4.  Area: square millimeter (mm$^2$).

Performance: Figure 3 plots performance vs. power consumption, with different labels for dataflow, neuromorphic, and PIM processors. The processors within a power consumption range of 1 W to 60 W have a performance of 1 to 275 TOPS. These are geared towards comparatively high-power applications such as surveillance systems, autonomous vehicles, industries, smart cities, and UAVs. The highest-throughput processors in this list are the EyeQ6 from MobileEye, the Journey 5 from Horizon, and the Jetson Orin from Nvidia. The Jetson Orin is about 2.15 times faster than both the EyeQ6 and Journey 5. From the company datasheet [153], the Jetson Orin has 275 TOPS at INT8 precision with 60 W of power. The Orin consumes about 1.5 and 2 times more power than the EyeQ6 and Journey 5, respectively. The processors with a power consumption of less than 1 W have a performance of 0.2 GOPS to 17 TOPS. These are targeted at edge and IoT applications. The least power is needed for PIM processors of the NDP series by Syntiant, which are flash-memory-based PIM processors [20].

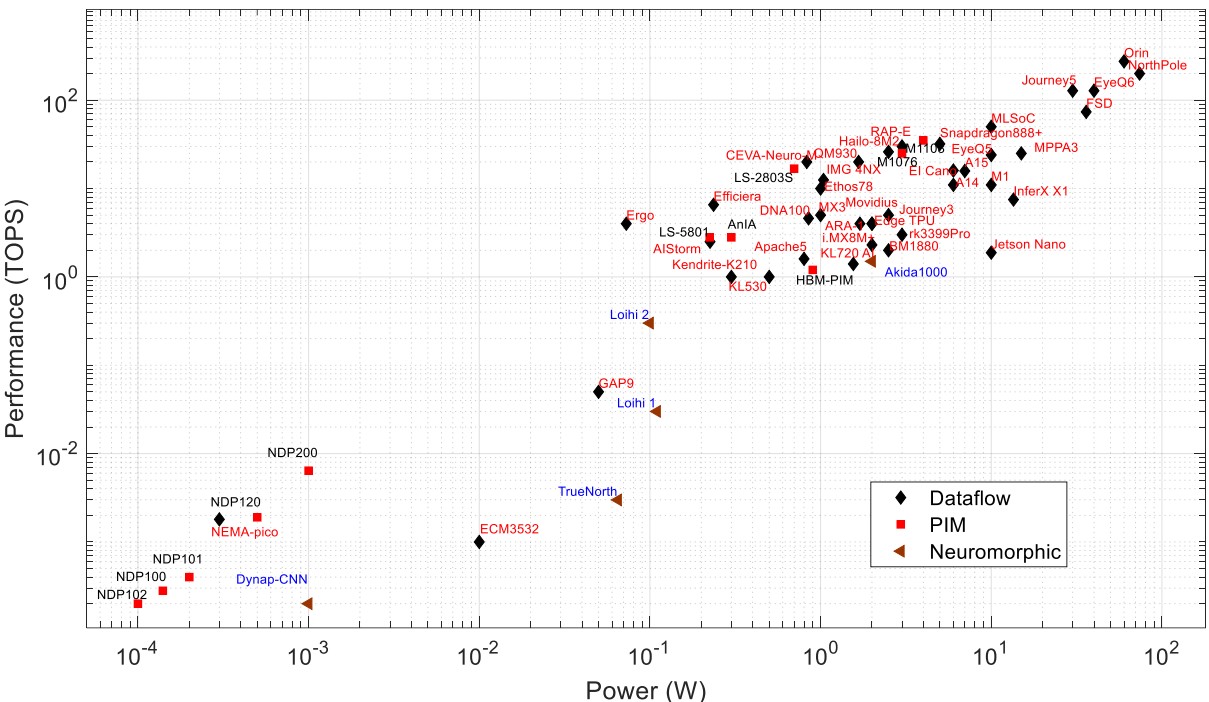

**Figure 3.** Power consumption and performance of AI edge processors.

The IBM NorthPole has 200 TOPS for INT8 precision at 60 W (based on a discussion with IBM). However, the NorthPole can have higher TOPS of 400 and 800 at 4 and 2 bit precision, respectively. According to a recent NorthPole article, the maximum power consumption of the NorthPole processor is 74 W [90].

Among neuromorphic processors, Loihi 2 outperforms other neuromorphic processors, except for the Akida AKD1000. The AKD1000, however, consumes 20× more power than the Loihi 2 (see Table 2). Although the neuromorphic processors seem less impressive in terms of TOPS vs. W, it is important to note that they generally need far fewer synaptic operations to perform a task if the task is performed with an algorithm that is natively spiking (i.e., not a deep network implemented with spiking neurons) [287].

The neuromorphic processors consume significantly less energy than other processors for inference tasks [227]. For example, the Loihi processor consumes 5.3× less energy than the Intel Movidious and 20.5× less energy than the Nvidia Jetson Nano [227]. Figure 3 shows that higher-performance PIM processors (such as the M1076, M1108, LS-2803S, and AnIA) exhibit similar computing speeds as dataflow or neuromorphic processors within the same range of power consumption (0.5 to 1.5 W).

Precision: Data precision is an important consideration when comparing processor performance. Figure 4 presents the precision of the processors from Figure 3. Figure 5 shows the distribution of precision and total number of processors for each architecture category. A processor may support more than one type of computing precision. Figures 3 and 4 are based on the highest precision supported by each processor.

Among dataflow processors, INT8 is the most widely supported precision for DNN computations. NVIDIA's Orin achieves 275 TOPS with INT8 precision, the maximum computing speed for INT8 precision in Figure 5. However, some processors utilize INT1 (Efficiara), INT64 (A15, A14, and M1), FP16 (ARA-1, DNA100, Jetson Nano, Snapdragon 888+), and INT16 (Ethos78, and Movidius). Neuromorphic and PIM processors mainly support INT1 to INT8 data precisions. Lower computing precisions generally reduce the inference accuracy. According to [236], VGG-9 and ResNet-18 have accuracy losses of 3.89% and 6.02%, respectively, for inference while computed with INT1 precision. A more in-depth discussion of the relationship between quantization and accuracy is presented in Section 3.1. A higher precision provides better accuracy but incurs more computing costs.

Figure 5 shows that the most common precision in the processors examined is INT8. This provides a good balance between accuracy and computational costs.

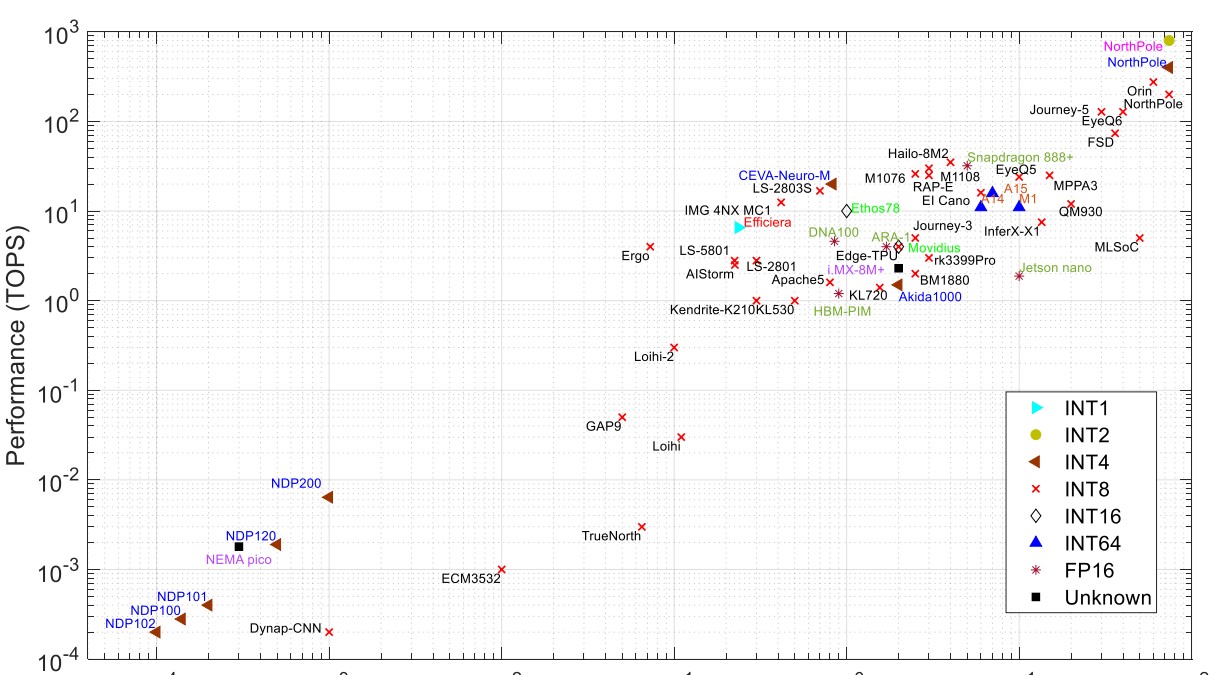

**Figure 4.** Power vs. performance of edge processors with computing precision.

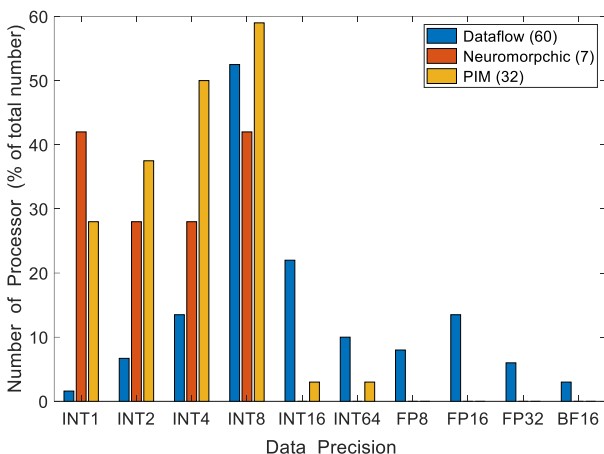

**Figure 5.** Number of edge processors supporting various degrees of data precision. The total number of processors is indicated in the legend.

As shown in Figures 4 and 5, almost all the neuromorphic processors use INT8 for synaptic computations. The exception to this is the AKD1000, which uses INT4 and shows the best performance among neuromorphic processors in terms of operations per second (1.5 TOPS). However, it consumes around 18× more power than Loihi processors. At INT8 precision, the Loihi 1 performs 30 GSOPS using 110 mW [138,223], whereas Loihi 2 surpasses this throughput by 10×, with a similar power consumption [9].

As shown in Figures 4 and 5, PIM processors primarily support precisions of INT1 to INT8. Figure 5 shows the performance of PIM processors in INT4 and INT8 precisions due to the unavailability of data for all supported precisions. Mythic processors (M1108 and M1076) manifest the best performance among PIM processors. Mythic and Syntiant have developed their PIM processors with flash memory devices. However, Mythic processors

require significantly higher power to compute DNNs in INT8 precision with its 76 computing tiles. Syntiant processors use INT4 precision and compute with about 13,000× lower throughput than Mythic M1076 while consuming about 6000× less power. The Syntiant processors are limited to smaller networks with up to 64 classes in NDP10x. On the other hand, Mythic processors can handle 10× more weights with greater precision [283]. The Samsung DRAM architecture-based PIM processor uses computing modules near the memory banks and supports INT64 precision [16].

Energy Efficiency: Figure 6 presents the performance vs. energy efficiency of dataflow for PIM and neuromorphic processors. Efficiency determines the computing throughput of a processor per watt. The energy efficiency of all PIM processors is located within 1 to 16 TOPS/W, whereas most of the dataflow processors are located in the 0.1 to 55 TOPS/W range. The PIM architecture reduces latency by executing the computation inside the memory modules, which increases computing performance and reduces power consumption. Loihi 2 manifests the best energy efficiency among all neuromorphic processors. Energy efficiency vs. power consumption, as shown in Figure 7, gives us a better understanding about the processors. Loihi 2 shows better energy efficiency than many high-performance edge AI processors, while it consumes very low power. Ergo is the most energy-efficient processor among all dataflow processors, which shows 55 TOPS/W.

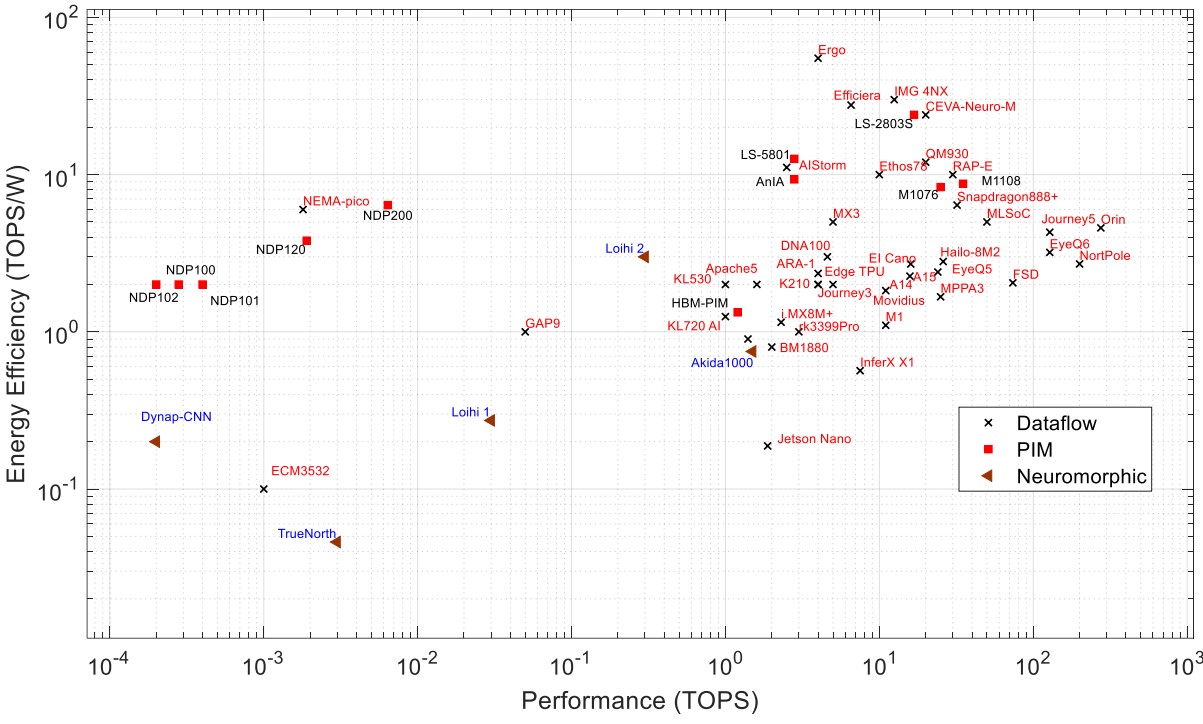

**Figure 6.** Performance and energy efficiency of edge processors.

Chip Area: The area is an important factor for choosing a processor for AI applications on edge devices. Modern processor technologies are pushing the boundaries to fabricate systems with very high density and superior performance at the same time. The smaller die area and lower power consumption is very important for battery-powered edge devices. The chip area is related to the cost of the silicon fabrication and also defines the application area. A smaller chip with high performance is desirable for edge applications.

Figures 7 and 8 present the power consumption and performance, respectively, vs. the chip area. It can be observed that in general, both the power consumption and performance increase with chip area. Based on the available chip sizes, the NothPole has the largest chip size of 800 mm$^2$ and performs 200 TOPS in INT8. The lowest-area chips have a dataflow architecture. Figure 9 shows the energy efficiency vs. area as the combined relationship of Figures 8 and 9. In this Figure, the PIM processors form a cluster. The overall energy

efficiency of this PIM cluster is higher than that of dataflow and neuromorphic processors of similar chip area. Some dataflow processors (such as Nema Pico, Efficiera, and IMG 4NX) exhibit higher energy efficiency and better performance vs. area than other processors.

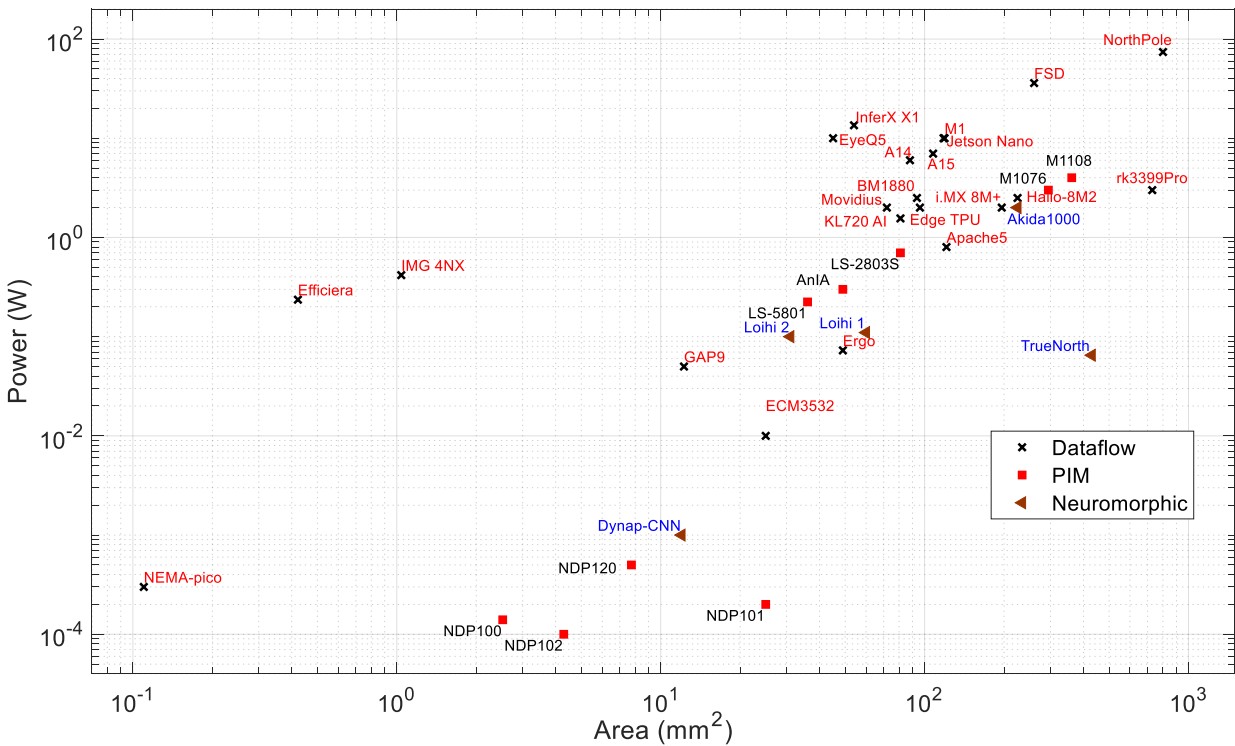

**Figure 7.** Power consumption vs. area of edge processors.

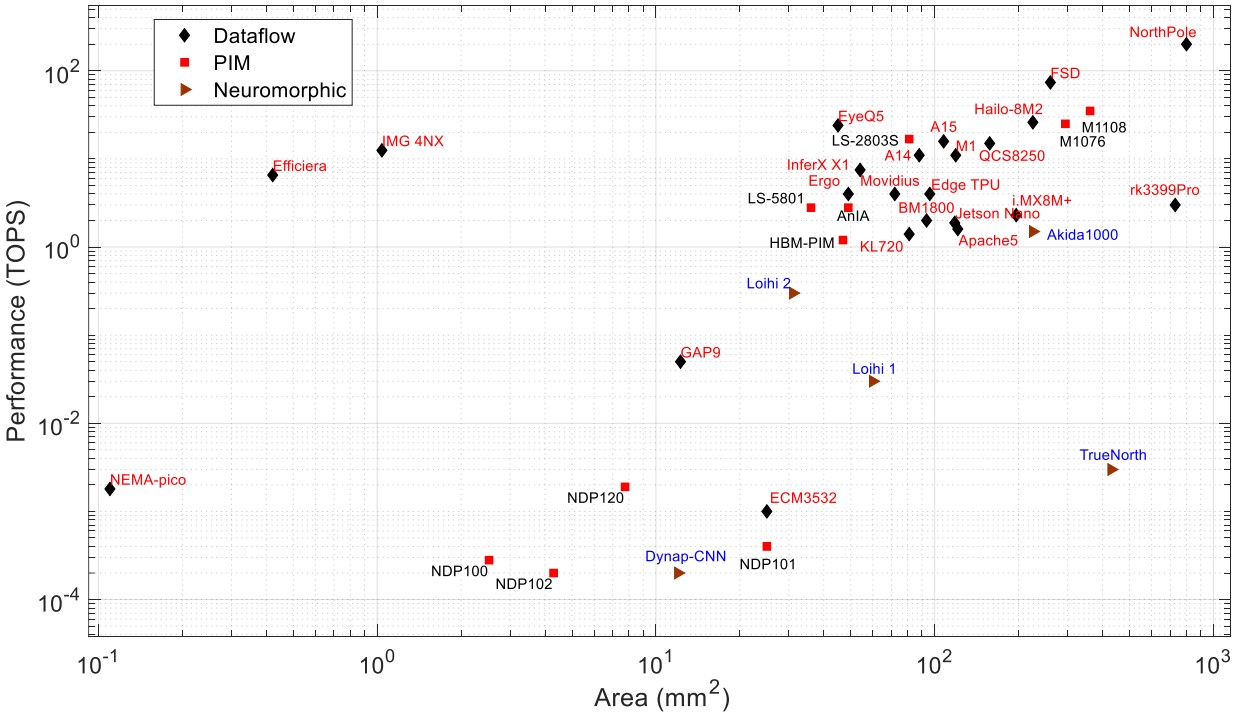

**Figure 8.** Area vs. performance of edge processors.

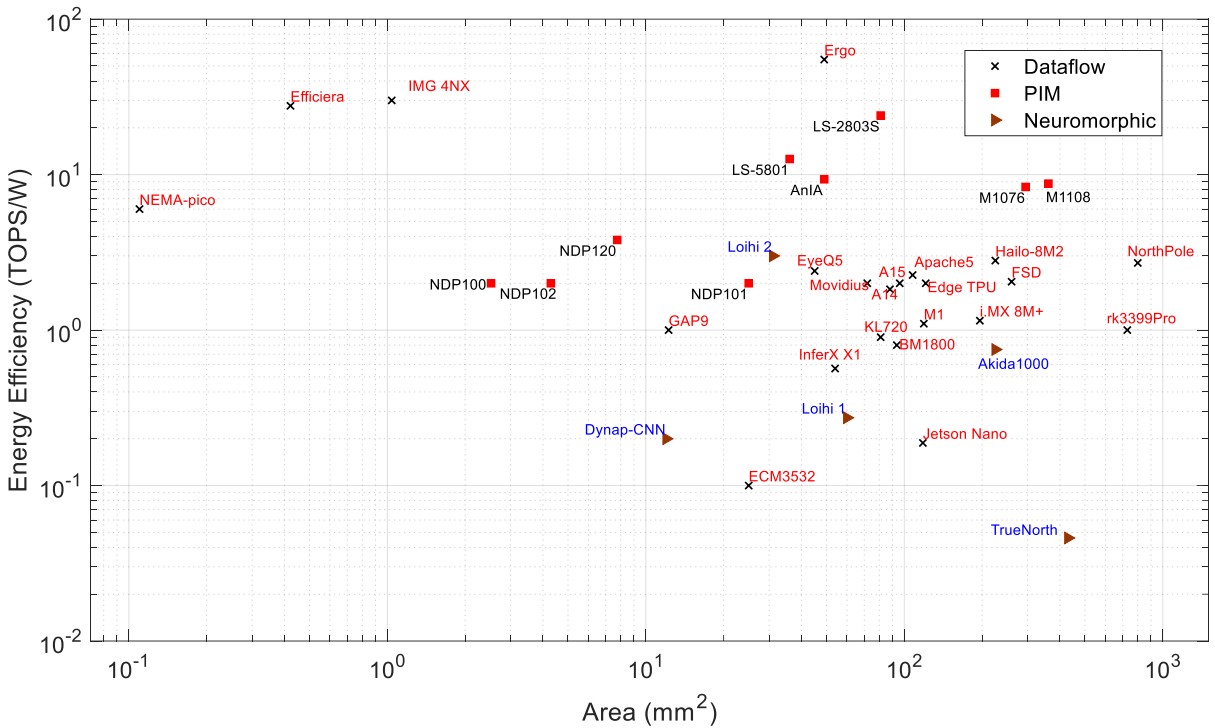

**Figure 9.** Area vs. energy efficiency of edge processors.

### 7.2. AI Edge Processors with PIM Architecture

While Figures 3–11 describe processors of all types, Figure 12 shows the relationship only between PIM processors that have either been announced as products or are still in industrial research. The research processors are presented in the conferences, such as ISSCC and VLSI. The PIM processors at the lower right corner of Figure 11 are candidates for data center and intensive computing applications [182–187]. PIM processors with higher energy efficiency are suitable for edge and IoT applications because of their smaller size, lower power consumption, and higher energy efficiency. From Figure 12, we can see that most of the PIM processors under industrial research show higher energy efficiency than already announced processors. This indicates that future PIM processors are likely to have much better performance and efficiency.

The PIM processors compute the MAC operation inside the memory array, thus reducing the data transfer latency. Generally, PIM processors compute in lower-integer/fixed-point precision. A PIM processor generally supports INT 1–16 precision. However, according to our study, we found around 59% of the PIM processors support INT8 precision for MAC operation, as shown in Figure 5. Low-precision computation is faster and requires lower power consumption compared to dataflow processors. PIM edge processors consume 0.0001 to 4 W for deep learning inference applications, as presented in Table 2 and Figure 3. However, the dataflow processors suffer from high memory requirements and latency issues, and they consume higher power than most of the PIM processors in order to achieve the same performance that we see in Figures 3–5.

From Figures 3 and 4, Syntient's NDP200 consumes less than 1 mW and shows the highest performance for extreme edge applications. Mythic M1108 consumes 4 W and exhibits the highest performance (35 TOPS) of all dataflow and neuromorphic processors that consume below 10 W of power. For the same chip area, the M1108 consumes 9× less power than Tesla's dataflow processor FSD, while FSD computes 2× faster than M1108, as presented in Figures 8 and 9.

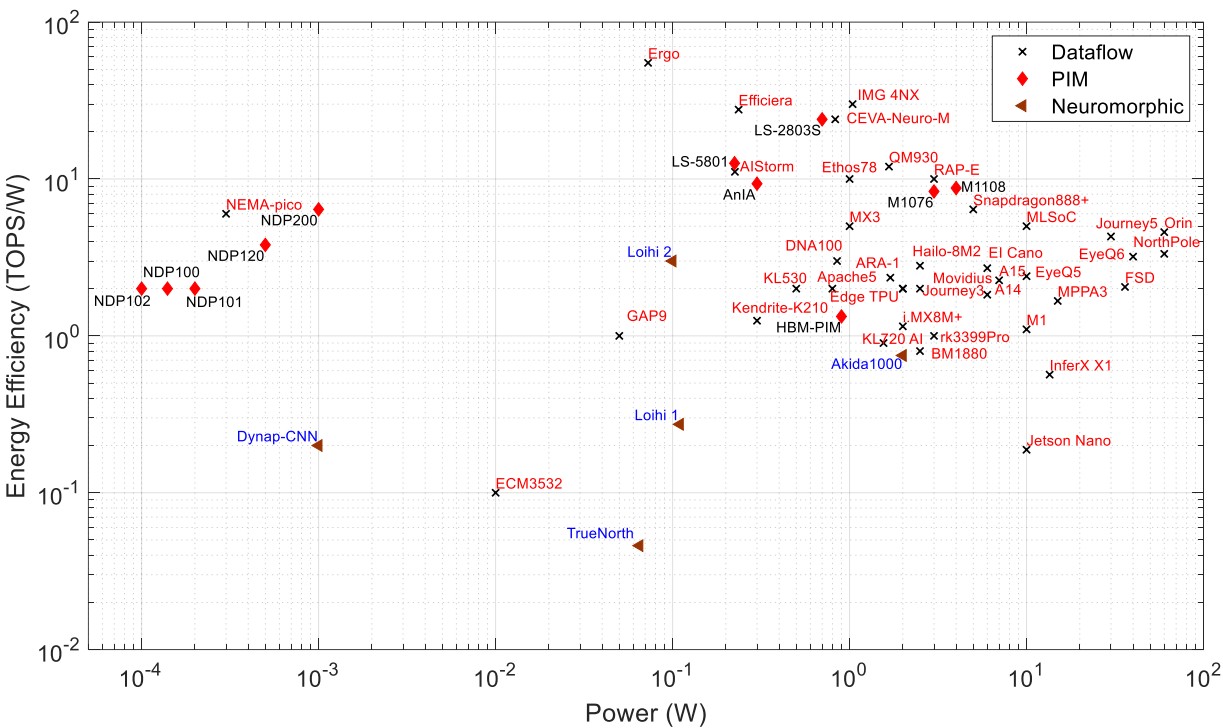

**Figure 10.** Power vs. energy efficiency of edge processors.

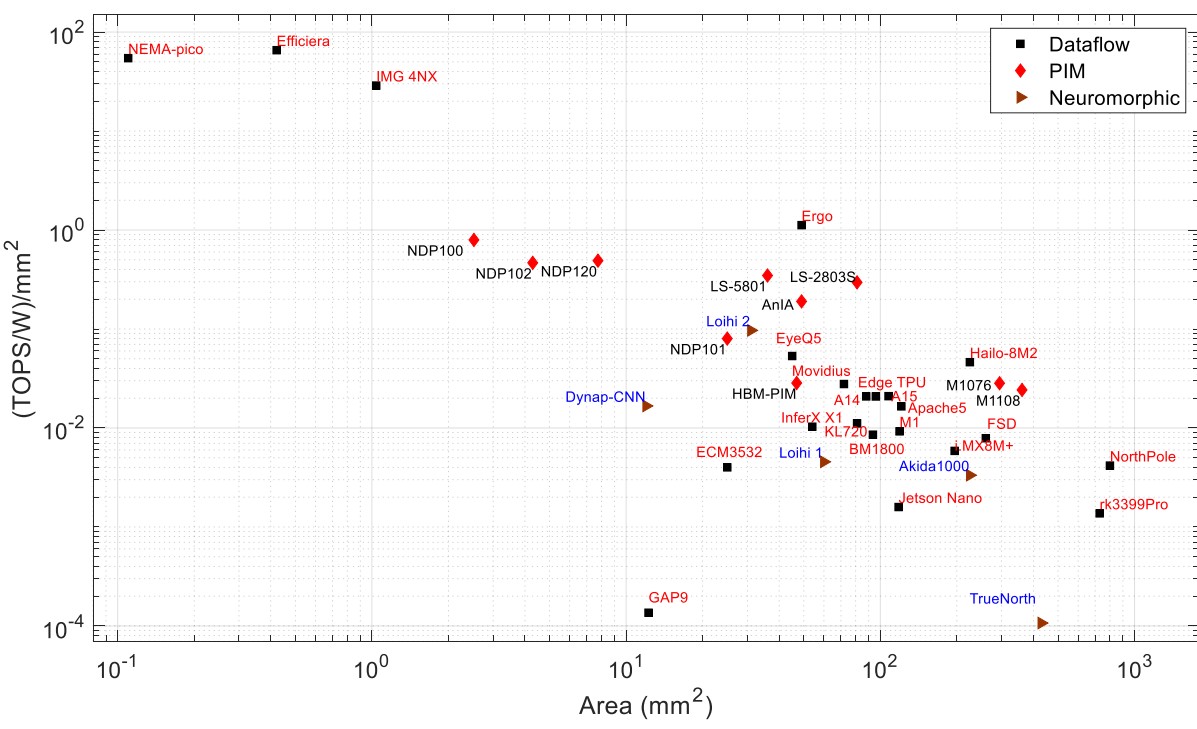

**Figure 11.** Area vs. energy efficiency per unit area of edge processors.

For the processors below 100 mm², Gyrfalcon's LS2803 shows the highest performance except for EyeQ5. However, EyeQ5 consumes about 14× higher power and performs 1.4× better than LS2803. The benefit of deploying PIM processors for edge applications is high performance with low power consumption, and the PIM processors reduce the computing latency significantly as the MAC operations are performed inside the memory array.

### 7.3. Edge Processors in Industrial Research

Several companies, along with their collaborators, are developing edge computing architectures and infrastructures with state-of-the-art performance. Figure 13 shows the power consumption vs. energy efficiency of the industrial research processors which were presented at high-tier conferences (such as ISSCC, VLSI). The chart includes both PIM [69,189,192,236–241,245–251,257,258,262,263] and dataflow [74,188,242–244,252–257, 259,261,264] processors.

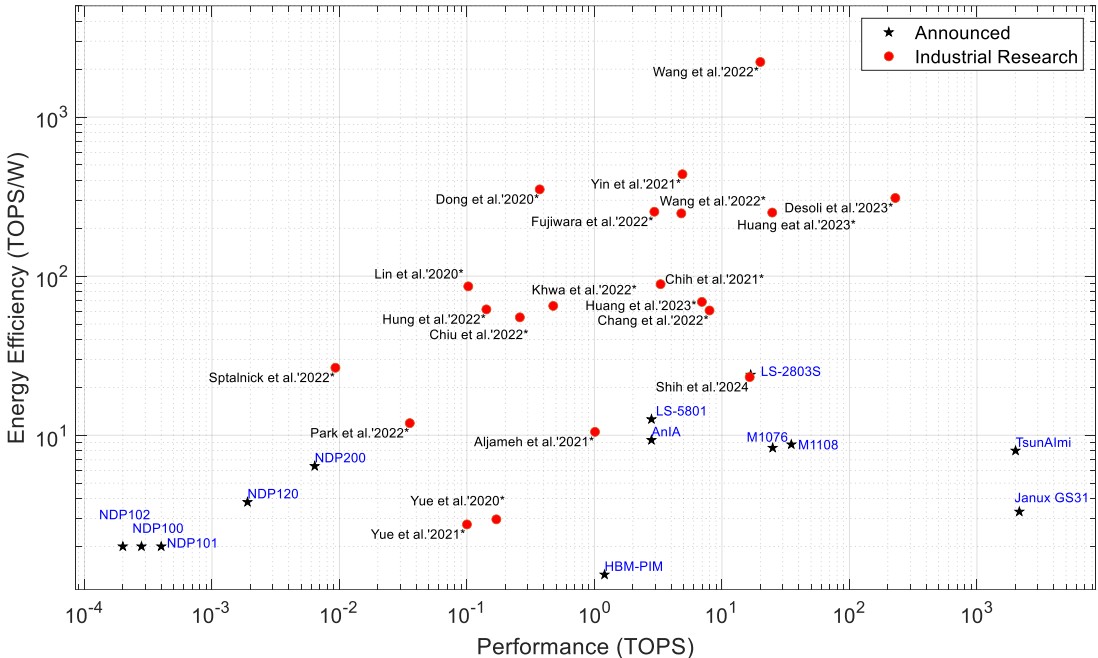

**Figure 12.** PIM (red) and dataflow (blue label) processors in industrial research. The references are used in this chart are [69,189,192,236–241,245–251,257,258,262,263] for PIM and [74,188,242–244, 252–257,259,261,264] for dataflow architecture.

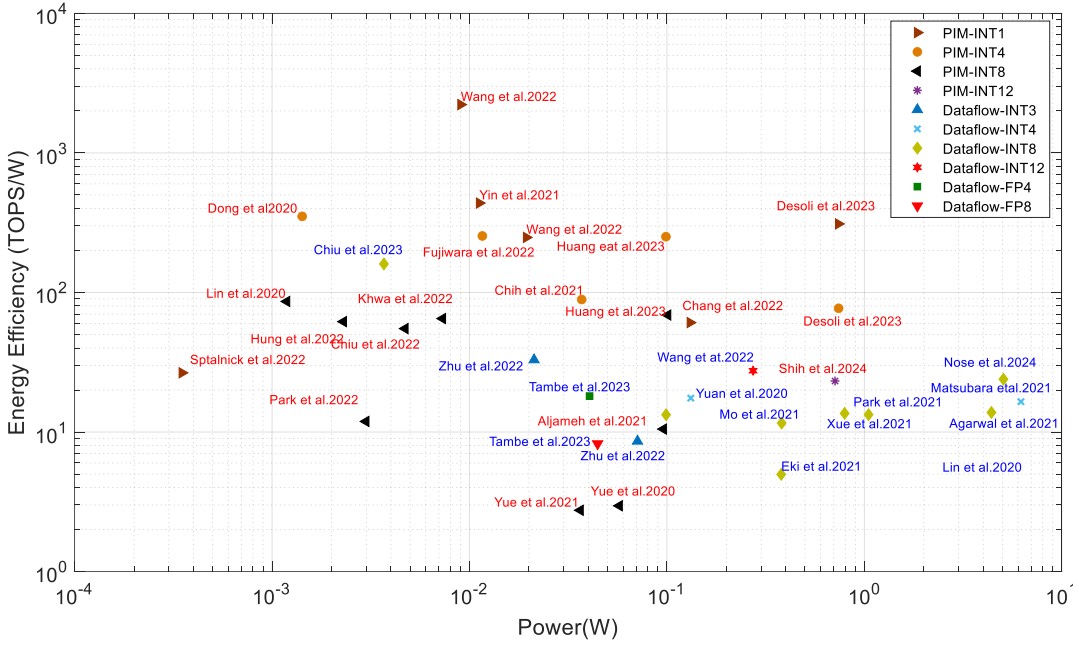

**Figure 13.** Performance vs. energy efficiency of PIM/CIM processors, Processors with an asterix (*) indicate the processors are still undergoing industrial research, and other processors have been released or announced by the manufacturer. For industrial PIM processors the examples are used from references [69,189,192,236–241,245–251,257,258,262,263].

Renesas Electronics presented a near-memory system in ISSCC 2024 developed in a 14 nm process that achieved 130.55 TOPS with 23.9 TOPS/W [264]. TSMC and National Tsing Hua University presented one near-memory system in a 22 nm CMOS process in ISSCC 2023 that computes 0.59 TOPS and 160 TOPS/W in 8-8-26-bit (input–weight–output) precision [260]. This system showed the highest energy efficiency amongst the near-memory and dataflow processors. The energy efficiency is achieved by a 90% input sparsity, while a 50% input sparsity gives an energy efficiency of 46.4 TOPS/W [260]. Alibaba and Fudan University [243] presented a processing near-memory system in ISSCC 2022 with 0.97 TOPS and 32.9 TOPS/W energy efficiency while computing with INT3 precision. This accelerator is a SRAM-based near-memory computing architecture. Tsing Microelectronics and Tsinghua University [252] demonstrated a dataflow processor in ISSCC 2022 for NLP and computer vision applications, which shows an energy efficiency of 27.5 TOPS/W in INT12 precision. Renesas Electronics [253] exhibited 13.8 TOPS/W in INT8 computing precision. Many other companies, such as IBM [254], Sony [256], MediaTek [257], and Samsung [242,255] have also demonstrated their research on dataflow edge processors with energy efficiencies around 11 to 18 TOPS/W.

PIM processors generally manifest better energy efficiencies than dataflow processors. TSMC and National Tsing Hua University presented a PIM system in a 16 nm CMOS process in ISSCC 2024 that achieved 98.5 TOPS/W in 8-8-23 precision (input–weight–output) [263]. MediaTek and TSMC presented a digital PIM system in ISCC 2024 developed in a 3 nm process that achieved 23.2 TOPS/W with 16.5 TOPS performance [262]. Intel and Columbia University demonstrated a PIM processor [249] in ISSCC 2022 that shows the performance and energy efficiency of 2219 TOPS/W and 20 TOPS, respectively, which is around 33× more efficient than the processor mentioned in [249].

However, the former processor uses far lower precision (INT1). TSMC and Tsinghua University [258] presented a PIM accelerator in ISSCC 2023 with 6.96 TOPS and 68.9 TOPS/W, which is about 12× faster than the near-memory computing system presented in [260] while computing in INT8 precision. STMicroelectronics presented a PIM accelerator that computes 57 TOPS and 77 TOPS/W in INT4 precision [261] that performs about 25× better than near-memory computing presented in ISSCC 2021 [188]. TSMC and Tsinghua University [246] presented a PCM-based processor in ISSCC 2022, which shows 65 TOPS/W in INT8 precision and is around 5× better than [260]. Samsung and Arizona State University [189] demonstrated PIMCA in VLSI' 2021 and showed an energy efficiency of 437 TOPS/W computed in INT1 precision. Other companies such as TSMC and collaborators [69,236,245–248,258,260], Samsung and collaborators [189], Intel and collaborators [188,241,249], and HK Hynix [192] have demonstrated their PIM processors at recent ISSCC and VLSI conferences.

### 7.4. Processor Selection, Price and Applications

This review is primarily focused on categorizing edge processors based on their underlying hardware architecture and computing techniques. The charts and tables reflect the different types of hardware architectures. Table 2 indicates the architecture types of the processors, such as dataflow, PIM or neuromorphic. However, it is also important to find the right processor for any application. Table 3 shows the manufacturer's suggested application areas of the processors. From the charts presented in Figures 3 and 4, the application fields of the processors can be determined based on the power consumption and performance metrics. Processors located in the lower left corner of Figures 3 and 4 are targeted at extreme edge applications such as wearable devices (smart watches, headphones, earbuds, and smart sunglasses). Syntiant's NDPs, Nema-Pico, DynapCNN processors are the candidate processors for these applications. The mid-range processors consume 0.1 to 10 W of power and are targeted at applications in security and surveillance. The processors include LS5801, DNA100, and CEVA-Neuro-M. The high-end processors are targeted at comparatively high-powered applications with about 100 TOPS computing performance. Target application areas include autonomous vehicles and industrial automation. The

candidate processors are Horizon's Journey series, Tesla's FSD, NVIDIA's Orin, Mobileye's EyeQ and IBM's NorthPole.

However, if we analyze the price of commercially available processors for edge applications, the prices vary based on the computing capability, energy efficiency and the types of applications. From this context, we can say that in general, the cost of a processor varies with performance (TOPS). The processors located in the lower left corner in Figures 3 and 4 exhibit the lowest performance and are in use in wearable AI devices that cost only a few dollars (USD 3–USD 10) [177]. The mid-range processors cost around USD 100, and the target applications are security and tracking applications. In this category, the Google Coral Edge TPU board costs USD 98 [288]. High-end edge processors can compute more than 100 TOPS and cost a few hundred to a couple of thousand dollars. These processors are mainly used in autonomous vehicles and in industry. For example, the current market price of the Tesla FSD is USD 8000 [289], and the NVIDIA Jetson Orin costs around USD 2000 [290].

## 8. Summary

This article reviewed different aspects and paradigms of AI edge processors released or announced recently by various tech companies. About 100 edge processors were examined. This work, however, did not cover DNN algorithms, HPC computing processors, or cloud computing. We categorized state-of-the-art edge processors and analyzed their performance, area, and energy efficiency to support the research community in edge computing. Multiple processing architectures including dataflow, neuromorphic, and PIM were examined. The performance and power consumption were analyzed for narrowing down edge AI processors for specific applications. Deep neural networks and software frameworks supported were discussed and are presented in tables.

Several of the edge processors offer on-chip retraining in real time. This enables the retraining of networks without having to send sensitive data to the cloud, thus increasing security and privacy. Intel's Loihi 2 and Brainchip's Akida processor can be retrained on local data for personalized applications and faster response rates.

This study found the power consumption and performance of processors varies in different architectures and application domains. For extreme wearable edge devices, power consumption ranges from 100 μW to a few mW, and computing throughput is around 1 GOPS. We found that many applications require higher computing performance, such as video processing and autonomous car operations. These high-performance applications consume a higher amount of power than extreme edge processors. For example, IBM's NorthPole computes at 200 TOPS with INT8 while consuming 60 W of power. This study found that for the same range of power consumption and chip size, PIM architectures perform better than dataflow or neuromorphic processors. This review found that the PIM processors show significant energy efficiency and consume less power compared to dataflow and neuromorphic processors. For example, the Mythic M1108 is a PIM processor and has the highest performance (35 TOPS), among dataflow and neuromorphic processors that consume less than 10 W of power. Neuromorphic processors are highly efficient for performing computation with less synaptic operations but may not be ideal for deep learning applications yet.

There are different types of deep learning frameworks for developing edge accelerators. The most common frameworks are TFL, ONNX, and Caffe2. Some providers developed their own framework to ease the development for users; for example, KaNN provides Kalray, and CEVA-DNN provides CEVA. Overall, TFL, Caffe2, and ONNX are the most popular platforms for developing DNN accelerator systems. Neuromorphic processors have different frameworks which support spike generation and computation, such as Nengo and Lava.

There are several emerging deep learning applications that are attracting significant interest. This includes generative AI models, such as transformer models used in ChatGPT and DALL-E for automated art generation. Transformer models are taking the AI world by

storm, as manifested by their super-intelligent chatbot and search queries. Generative AI models also now have a place in image and creative art generation. Transformer engines are mainly designed for data centers or cloud applications, but some processors, such as NVIDIA Hopper H100 [291], can be used for edge workloads. Samsung has released digital PIM for generative AI applications in the data center and edge [292]. ResNet, GoogleNet, and YOLO models are also being used in various industries for facial recognition, lane keeping assistance, and surveillance. Deep reinforcement learning is becoming popular for autonomous learning models in dynamic environments. All of these applications could benefit from highly efficient specialized processors that could run the applications locally, without the need for cloud access. Future directions for industry could be to implement these algorithms in emerging non-von Neumann computing paradigms for low-power computing on edge devices. Current dataflow processors, such as the NVIDIA Orin or the IBM NorthPole, would probably be able to handle these applications without any changes. More emerging architectures, such as PIM and neuromorphic technologies, may need more enhancements to enable these applications to run on edge devices.

**Author Contributions:** S.A. and T.M.T. are the main contributors. They conceptualized the review idea and collected data, visualized and critically analyzed hardware performance. C.Y., Q.W., M.B. and S.K. have contributed equally to review and editing. All authors have read and agreed to the published version of the manuscript.

**Funding:** Funding is provided by the Department of Electrical and Computer Engineering, University of Dayton, Dayton, OH 45469, USA.

**Conflicts of Interest:** The authors declare no conflict of interest.

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
