# Peer review of "Survey of Deep Learning Accelerators for Edge and Emerging Computing"

_electronics, doi:10.3390/electronics13152988_

Round 1

Reviewer 1 Report

Comments and Suggestions for Authors

This paper provides a comprehensive review of AI devices on the edge. It reviews the tools used, the types of devices and gives a characterization in performance, energy efficiency, power and area.

From the point of view of a designer of embedded systems in a very useful and very well focused work.

It would be perfect if they included two fundamental aspects, the cost of the devices and an assessment (albeit subjective) of the ease of development according to the frameworks.

With this study I am very surprised that MAX78000 devices from Analog Devices [https://www.analog.com/en/products/max78000.html] have not been studied, I think it is mandatory to include it.

Typing error:

ii. Ditection -> ii. Detection

Author Response

Comments 1: This paper provides a comprehensive review of AI devices on the edge. It reviews the tools used, the types of devices and gives a characterization in performance, energy efficiency, power and area.

From the point of view of a designer of embedded systems in a very useful and very well focused work.

Response 1: Thank you.

Comments 2: It would be perfect if they included two fundamental aspects, the cost of the devices and an assessment (albeit subjective) of the ease of development according to the frameworks.

Response 2: We agree. To emphasize this point, we have added the paragraphs below. Please keep in mind that finding the price for all processors is not possible. Additionally, many processors are used for company products, and not for purchase in the market. Some processors are still used only for research purposes. The paragraph on cost can be found in Page 35, lines 987-997. Comments on frameworks is added in the summary section, Page 36, lines 1027-1033.

“However, if we analyze the price of commercially available processors for edge applications, the prices vary based on the computing capability, energy efficiency and the types of applications. From this context, we can say in general the cost of a processor varies with performance (TOPS). The processors located in the lower left corner in Fig. (3,4) exhibit the lowest performance and are in use in wearable AI devices that cost only a few dollars ($3-$10) [314,315]. The mid-range processors cost around $100, and the target applications are security and tracking applications. In this category, the Google Coral Edge TPU board costs $98 [316]. High-end edge processors can compute more than 100 TOPS and cost a few hundred to a couple of thousand dollars. These processors are mainly used in autonomous vehicles and in industry. For example, the current market price of the Tesla FSD is $8,000 [317], and the NVIDIA Jetson Orin is around $2,000 [318].”

“There are different types of deep learning frameworks for developing edge accelerators. The most common frameworks are TFL, ONNX, and Caffe2. Some providers developed their own framework to ease the development for the developers, such as KaNN provides Kalray, CEVA-DNN provides CEVA. Overall, TFL, Caffe2, and ONNX are the most popular platforms to develop the DNN accelerator systems. Neuromorphic processors have different frameworks which support spike generation and computation, such as Nengo and Lava.”

Comments 3: With this study I am very surprised that MAX78000 devices from Analog Devices [https://www.analog.com/en/products/max78000.html] have not been studied, I think it is mandatory to include it.

Response 3: We agree. We have added the processor in the text but could not update the chart as we did not find the exact power consumption and performance (TOPS) metrics in the suggested link and other links that we had gone through. The changes on Comment three can be found on Table II, row-1, Page 13, and page 11, lines 394-405.

“Analog Devices Inc. developed a low cost mixed-signal CNN accelerator MAX7800 that consists of a Cortex-M4 processor, a 32-bit RISC-V processor with Floating Point Unit (FPU) and CPU for system control with a DNN accelerator. The accelerator has a SRAM based 442 KB on-chip weight storage memory which can support 1, 2, 4, and 8 bit weights. The CNN engine has 64 parallel processors and 512 KB data memory. Each processor has a pooling unit and a convolutional unit with a dedicated memory unit. The processor consumes 1pJ/MAC operation. As the exact power consumption (W) and performance (TOPS) data is not publicly available at the time of this writing, we did not add it to our graphs. The size of the chip is 64 mm2. The architecture supports Pytorch, and Tensorflow toolsets for the development of a range of DNN models. The target application areas are object detection, classification, face recognition, time series data processing, and noise cancellation.”

Comments 4: Typing error:

ii. Ditection -> ii. Detection

Response 4: Thank you for catching this error.

We have Corrected this typo. Page 5, line 173.

4. Response to Comments on the Quality of English Language

Point 1:

Response 1:    No Comments

5. Additional clarifications

All Updates are highlighted in yellow on the word file in the paper.

Reviewer 2 Report

Comments and Suggestions for Authors

The paper is an almost exhaustive survey on processors that can be used for edge computing and have support for AI methods' implementation. The paper contains an impressive presentation of approximately 100 processors of different classes, types and made by different more or less known producers.

Strengths: the authors properly identified the types of processors that can be used as candidates of edge processors in applications that require AI models and their inference.

- The multitude of analyzed processors

- graphical evaluation and comparison of different processors

Weaknesses: it is difficult to follow all the processor variations;

- the authors does not group the processors based on some application types or classes; at the end of the reading it is difficult to decide which processor would be better for a given type of application

- even if it is difficult to determine, some general tendencies and future trends would be useful for the reader

- maybe a selection of some representative processors from each class would make the reader's job easier, and the length of the paper would be shorter

- on lines 74 and 688 the lines are not split properly

Overall the paper is a good reference for those who are trying to decide which processor or platform would be a better choice for their implementation.

Author Response

Comments 1: The paper is an almost exhaustive survey on processors that can be used for edge computing and have support for AI methods' implementation. The paper contains an impressive presentation of approximately 100 processors of different classes, types and made by different more or less known producers.

Response 1: Thank you.

Comments 2: Strengths: the authors properly identified the types of processors that can be used as candidates of edge processors in applications that require AI models and their inference.

- The multitude of analyzed processors

- graphical evaluation and comparison of different processors

Response 2: Thank you.

Comments 3: Weaknesses: it is difficult to follow all the processor variations;

- the authors does not group the processors based on some application types or classes; at the end of the reading it is difficult to decide which processor would be better for a given type of application

- even if it is difficult to determine, some general tendencies and future trends would be useful for the reader

- maybe a selection of some representative processors from each class would make the reader's job easier, and the length of the paper would be shorter

Response 3: We agree. Although we did not organize the processors based on applications, the processors are organized based on the underlying architecture, such as Dataflow, Neuromorphic and Computing in-Memory. However, based on the suggestions, we have added the paragraph below in the text on the application domain and how to find the processors for a particular application field. A paragraph is added in Page 35, line 969-986.

“This review is primarily focused on categorizing edge processors based on their underlying hardware architecture and computing techniques. The charts and tables reflect the different types of hardware architecture. Table II indicates the architecture types of the processors, such as dataflow, PIM or neuromorphic. However, it is also important to find the right processor for any application. Table III shows the manufacturer suggested application areas of the processors. From the charts presented in Figs. 3 and 4, the application fields of the processors can be determined based on the power consumption and performance metrics. Processors located in the lower-left corner of Figs. 3 and 4 are targeted for extreme edge applications such as wearable devices (smart watches, headphones, earbuds, and smart sunglasses). Syntiant’s NDPs, Nema-Pico, DynapCNN processors are the candidate processors for these applications. The mid-range processors consume 0.1 to 10W of power and target applications in security and surveillance. The processors include LS5801, DNA100, and CEVA-Neuro-M. The high-end processors are targeted for comparatively high-powered applications with about 100 TOPS computing performance. Target application areas include autonomous vehicles and industrial automations. The candidate processors are Horizon’s Journey series, Tesla’s FSD, NVIDIA’s Orin, Mobileye’s EyeQ and IBM’s NorthPole..”

Comments 4:

- on lines 74 and 688 the lines are not split properly

Response 4: We agree. We have corrected this issue, page 3, line 73-76, and Page 23, line 703-705.

Comments 5:

Overall the paper is a good reference for those who are trying to decide which processor or platform would be a better choice for their implementation.

Response 5:  Thank you for your evaluation.

4. Response to Comments on the Quality of English Language

Point 1:

Response 1:   No Comments

5. Additional clarifications

All Changes are highlighted in yellow in the main text.
